# Flipped Classroom:
# Effective Teaching for Time Series Forecasting

**Philipp Teutsch** *philipp.teutsch@tu-ilmenau.de*
*Technische Universität Ilmenau*

**Patrick Mäder** *patrick.maeder@tu-ilmenau.de*
*Technische Universität Ilmenau*
*Friedrich-Schiller-Universität Jena*

**Reviewed on OpenReview:** *https://openreview.net/forum?id=w3x20YEcQK*

## Abstract

Sequence-to-sequence models based on LSTM and GRU are a most popular choice for forecasting time series data reaching state-of-the-art performance. Training such models can be delicate though. The two most common training strategies within this context are teacher forcing (TF) and free running (FR). TF can be used to help the model to converge faster but may provoke an exposure bias issue due to a discrepancy between training and inference phase. FR helps to avoid this but does not necessarily lead to better results, since it tends to make the training slow and unstable instead. Scheduled sampling was the first approach tackling these issues by picking the best from both worlds and combining it into a curriculum learning (CL) strategy. Although scheduled sampling seems to be a convincing alternative to FR and TF, we found that, even if parametrized carefully, scheduled sampling may lead to premature termination of the training when applied for time series forecasting. To mitigate the problems of the above approaches we formalize CL strategies along the training as well as the training iteration scale. We propose several new curricula, and systematically evaluate their performance in two experimental sets. For our experiments, we utilize six datasets generated from prominent chaotic systems. We found that the newly proposed increasing training scale curricula with a probabilistic iteration scale curriculum consistently outperforms previous training strategies yielding an NRMSE improvement of up to 81% over FR or TF training. For some datasets we additionally observe a reduced number of training iterations. We observed that all models trained with the new curricula yield higher prediction stability allowing for longer prediction horizons.

## 1 Introduction

Advanced Recurrent Neural Networks (RNNs) such as Long Short Term Memory (LSTM) (Hochreiter & Schmidhuber, 1997) and Gated Recurrent Unit (GRU) (Cho et al., 2014) achieved significant results in predicting sequential data (Chung et al., 2014; Nowak et al., 2017; Yin et al., 2017). Such sequential data can for example be textual data as processed for Natural Language Processing (NLP) tasks where RNN models were the method of choice for a long time, before feed-forward architectures like transformers showed superior results in processing natural language data (Devlin et al., 2018; Yang et al., 2019; Radford et al., 2019; Brown et al., 2020). Shifting the view to the field of modeling dynamical or even chaotic systems, encoder-decoder RNNs are still the method of choice for forecasting such continuous time series data (Wang et al., 2019; Thavarajah et al., 2021; Vlachas et al., 2018; Sehovac et al., 2019; Sehovac & Grolinger, 2020; Shen et al., 2020; Walther et al., 2022; Pandey et al., 2022).

Nevertheless encoder-decoder RNNs do have their difficulties, especially when it comes to effectively training them. Firstly, there is the exposure bias issue that can appear when teacher forcing (TF) is used for training

the model. TF is the strategy that is typically applied when training RNNs for time series sequence-to-sequence tasks (Williams & Zipser, 1989), regardless of the type of data. To understand the problem, we first give a quick side note about the motivation behind TF. Its main advantage is that it can significantly reduce the number of steps a model needs to converge during training and improve its stability (Miao et al., 2020). However, TF may result in worse model generalization due to a discrepancy between training and testing data distribution. It is less resilient against self-induced perturbations caused by prediction errors in the inference phase (Sangiorgio & Dercole, 2020). However, simply training the model without TF in free running (FR) mode instead does not necessarily provide convincing results, as can be seen in Section 5.6. Rather a more sophisticated strategy is needed to guide the model through the training procedure.

Several authors propose methods to mitigate the exposure bias or improve training stability and results in general inventing certain strategies (Bengio et al., 2015; Nicolai & Silfverberg, 2020; Lamb et al., 2016; Liu et al., 2020; Dou et al., 2019; Hofmann & Mäder, 2021). Even though most of these methods address the training for NLP tasks, such as text translation, text completion or image captioning, we focus on time series forecasting. Our datasets consist of sequences sampled from approximations of different chaotic systems. With these systems, the next state can always be derived from the past state(s) deterministically, but they also tend to be easily irritated by small perturbations. Thus the trained model needs to be especially resilient against small perturbations when auto-regressively predicting future values, otherwise those small errors will quickly accumulate to larger errors (Sangiorgio & Dercole, 2020). These aspects make the kind of data we use especially challenging. Since TF avoids the error accumulation only during training, we argue the models are even more vulnerable to issues like the exposure bias and will thus more likely fail to forecast larger horizons.

Around the interest of predicting particularly (chaotic) dynamical systems with RNNs, another field has formed that very intensively studies several specialized RNN-based approaches proposed to stabilize the training process and prevent exploding gradients. Most of propositions include architectural tweaks or even new RNN architectures considering the specifics of dynamical systems and their theory (Lusch et al., 2018; Vlachas et al., 2018; Schmidt et al., 2019; Champion et al., 2019; Chang et al., 2019; Rusch & Mishra, 2020; Erichson et al., 2020; Rusch et al., 2021; Li et al., 2021).

Monfared et al. (2021), for example, performed a theoretical analysis relating RNN dynamics to loss gradients and argue that this analysis is especially insightful for chaotic systems. With this in mind they suggest a kind of sparse teacher forcing (STF), inspired by the work of Williams & Zipser (1989) that uses information about the degree of chaos of the treated dynamical system. As a result, they form a training strategy that is applicable without any architectural adaptations and without further hyper-parameters. Their results using a vanilla RNN, a piecewise linear recurrent neural network (PLRNN) and an LSTM for the Lorenz (Lorenz, 1963) and the Rössler (Rössler, 1976) systems show clear superiority of applying chaos-dependent STF.

Reservoir computing RNNs were successfully applied to chaotic system forecasting and analysis tasks as well. For example, Pathak et al. (2017) propose a reservoir computing approach that fits the attractor of chaotic systems and predicts their Lyapunov exponents.

In this paper, we focus on CL training strategies like scheduled sampling that require no architectural changes of the model and thus can be applied easily for different and existing sequence-to-sequence (seq2seq) models. All presented strategies will be evaluated across different benchmark datasets. Our main contributions are the following: First, assembling a set of training strategies for encoder-decoder RNNs that can be applied for existing seq2seq models without adapting their architecture. Second, presenting a collection of strategies' hyper-parameter configurations that optimize the performance of the trained model. Third, proposing a "flipped classroom" like strategy that outperforms all existing comparable approaches on several datasets sampled from different chaotic systems. Fourth, proposing a method that yields substantially better prediction stability and therefore allows for forecasting longer horizons.

The course of the paper continues with Section 2, where we provide the background of sequence-to-sequence RNNs and the conventional ways to train them. We also give a short introduction to chaotic behavior of our data. In Section 3 we examine existing approaches dealing with the difficulty of training seq2seq RNNs in the context of different applications. Also, we give an overview of work treating RNNs for chaotic system prediction in particular. Section 4 describes how we designed our training strategies and how they

are applied. Further information about the experimental setup and our results we present in Section 5. In Section 6 we discuss the results, the strengths and limitations of the different strategies. Finally, in Section 7, we conclude our findings.

## 2 Background

Within this paper, we study Curriculum Learning (CL) training strategies for sequence-to-sequence RNNs. We are using an encoder-decoder architecture (Chung et al., 2014) as model to forecast time series data. The rough structure of the encoder-decoder architecture is shown in Figure 1a. It consists of two separate RNNs, an encoder and a decoder. The encoder is trained to build up a hidden state representing the recent history of the processed time series. It takes an input sequence $(x_1, x_2, \ldots, x_n)$ of $n$ input values where each value $x_j \in \mathbb{R}^d$ is a $d$ dimensional vector. For each processed step, the encoder updates its hidden state to provide context information for the following steps. The last encoder hidden state (after processing $x_n$) is then used as initial hidden state of the decoder. Triggered by a sequence's preceding value, the decoder's task is predicting its next future value while taking into account the sequence's history via the accumulated hidden state. For a trained network in the inference phase, that means that the decoder's preceding prediction is auto-regressively fed back as input into the decoder (aka auto-regressive prediction) (cp. Fig. 1a). All $m$ outputs $y_j \in \mathbb{R}^d$ together form the output sequence $(y_1, y_2, \ldots, y_m)$.

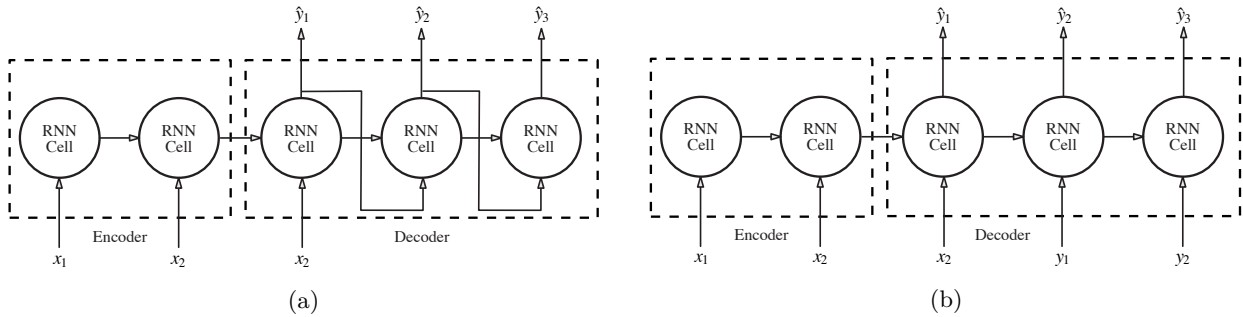

Figure 1: RNN encoder-decoder architecture in inference phase (a) and in training phase using teacher forcing (b)

### 2.1 Training Sequence Prediction Tasks

While training, the decoder's inputs may either be previously predicted outputs (free running) or known previous values stemming from the given dataset (teacher forcing). Training in **free running** mode might be an intuitive approach, but slows down the training. This is due to accumulated error throughout multiple prediction steps without the support of teacher forced inputs – especially in the early training epochs (Nicolai & Silfverberg, 2020). In contrast, **teacher forcing** aims to avoid this problem by utilizing the corresponding previous ground truth value as decoder input rather than the previously predicted value. This way, the model learns from the beginning of the training on to adapt its weights to perfect input values and converges noticeable faster (Miao et al., 2020) (cp. Fig. 1b). However,TF also bears a significant drawback, since during training the model is never exposed to the noisy predicted values it will later face in the inference phase. Therefore does often not generalize very well. A model trained with TF solely learned to predict on basis of perfect values and is thus vulnerable for small perturbations on its input, this effect is called **exposure bias** (Ranzato et al., 2015).

### 2.2 Chaotic Systems

The data we are using for our experiments later in Section 5 is generated from chaotic systems. Whether or not a dynamical system is chaotic, can be confirmed by considering its Lyapunov exponents $\lambda_k$ for $k \in [1, d]$. Given an initial perturbation $\varepsilon_0$ the exponential rate with which the perturbation will increase (or decrease) in the direction of dimension $i$ is the Lyapunov exponent $\lambda_k = \lim_{t \to \infty} \frac{1}{t} \ln(\frac{||\varepsilon_t||}{||\varepsilon_0||})$ (Dingwell, 2006). That

is, the Lyapunov exponents denote how sensitive the system is to the initial conditions (initial state). A deterministic dynamical system with at least one positive Lyapunov exponent while being aperiodic in its asymptotic limit is called **chaotic** (Dingwell, 2006). Analogously, dynamical systems with at least two positive, one negative, and one zero Lyapunov exponent are called **hyper-chaotic** systems. Dingwell points out that the largest Lyapunov exponent (LLE) can be used as a measure to compare the chaotic behavior of different systems.

## 3 Related Work

Schmidt (2019) defines exposure bias as describing "a lack of generalization with respect to an – usually implicit and potentially task and domain dependent – measure other than maximum-likelihood" meaning that when the exclusive objective is to maximize the likelihood between the output and the target sequence, one can use TF during training (Goodfellow et al., 2016). However, Goodfellow et al. argue that the kind of input the model sees while testing will typically diverge from the training data and the trained model may lack the ability of correcting its own mistakes. Thus, in practice, the TF can be a proper training strategy but may hinder the model to learn compensating its inaccuracies. He et al. study exposure bias for natural language generation tasks (He et al., 2019). They use sequence- and word-level quantification metrics to observe the influence of diverging prefix distributions on the distribution of the generated sequences. Two distributions are generated. One with and one without induced exposure bias. Those two distributions are then compared on basis of the corresponding corpus-bleu scores (Papineni et al., 2002). The study concludes that for language generation, the effect of the exposure bias is less serious than widely believed.

Several studies propose approaches to overcome the exposure bias induced by TF. The earliest of these studies proposes **scheduled sampling** (Bengio et al., 2015). Scheduled sampling tries to take the advantages of training with TF while also acclimating the trained model to its own generated data error. It does that by using ground truth values as input for a subset of the training steps and predicted values for the remaining. $\epsilon_i$ denotes the TF probability at step $i$. Accordingly, the probability of using the predicted value is $1 - \epsilon_i$. During training $\epsilon_i$ decreases from $\epsilon_s$ to $\epsilon_e$. This procedure changes the input data while training, making it a curriculum learning approach, as which scheduled sampling was proposed and works without major architectural adaptions. Originally proposed for image captioning tasks, scheduled sampling was also applied, e.g., for sound event detection (Drossos et al., 2019). Nicolai and Silfverberg consider and study the TF probability $\epsilon$ as a hyper-parameter. Rather than using a decay function that determines the decrease of $\epsilon_i$ over the course of training epochs, they use a fix TF probability throughout the training (Nicolai & Silfverberg, 2020). They observed a moderate improvement compared to strict TF training. Scheduled sampling is not restricted to RNN-based sequence-to-sequence models though, it has also been studied for transformer architectures (Mihaylova & Martins, 2019). Mihaylova and Martins tested their modified transformer on two translation tasks but could only observe improved test results for one of them.

Apart from scheduled sampling (Bengio et al., 2015), a number of approaches have been proposed typically aiming to mitigate the exposure bias problem by adapting model architectures beyond an encoder-decoder design. **Professor forcing** (Lamb et al., 2016) is one of these more interfering approaches that aims to guide the teacher forced model in training by embedding it into a Generative Adversarial Network (GAN) framework (Goodfellow et al., 2014). This framework consists of two encoder-decoder RNNs that respectively form the generator and the discriminator. The generating RNNs have shared weights that are trained with the same target sequence using their respective inputs while at the same time they try to fool the discriminator by keeping their hidden states and outputs as similar as possible. The authors conclude that their method, compared to TF, provides better generalization for single and multi-step prediction. In the field of Text-To-Speech (TTS), the concept of professor forcing has also been applied in the GAN-based training algorithm proposed by Guo et al. (2019). They adapted professor forcing and found that replacing the TF generator with one that uses scheduled sampling improved the results of their TTS model in terms of intelligibility. As another approach for TTS, (Liu et al., 2020) proposed **teacher-student training** using a training scheme to keep the hidden states of the model in FR mode close to those of a model that was trained with TF. It applies a compound objective function to align the states of the teacher and the student model. The authors observe improved naturalness and robustness of the synthesized speech compared to their baseline.

Dou et al. (Dou et al., 2019) proposed **attention forcing** as yet another training strategy for sequence-to-sequence models relying on an attention mechanism that forces a reference attention alignment while training the model without TF. They studied TTS tasks and observed a significant gain in quality of the generated speech. The authors conclude that attention forcing is especially robust in cases where the order of predicted output is irrelevant.

The discussed approaches for mitigating exposure bias were proposed in the context of NLP and mainly target speech or text generation. For time series forecasting, Sangiorgio & Dercole (2020) suggest to neglect TF completely and solely train the model in FR mode, thereby, sacrificing the faster convergence of TF and potentially not reaching convergence at all.

In the context of RNNs for forecasting and analyzing dynamical systems, the majority of existing work deals with exploding and vanishing gradients as well as capturing long-term dependencies while preserving the expressiveness of the network. Various studies rely on methods from dynamical systems theory applied to RNN or propose new network architectures.

Lusch et al. (2018) and Champion et al. (2019) use a modified autoencoder to learn appropriate eigenfunctions that the Koopman operator needs to linearize the nonlinear dynamics of the system. In another study, Vlachas et al. (2018) extend an LSTM model with a mean stochastic model to keep its state in the statistical steady state and prevent it from escaping the system's attractor. Schmidt et al. (2019) propose a more generalized version of a PLRNN (Koppe et al., 2019) by utilizing a subset of regularized memory units that hold information much longer and can thus keep track of long-term dependencies while the remaining parts of the architecture are designated to approximate the fast-scale dynamics of the underlying dynamical system. The Antisymmetric Recurrent Neural Network (AntisymmetricRNN) introduced by Chang et al. (2019) represents an RNN designed to inherit the stability properties of the underlying ordinary differential equation (ODE), ensuring trainability of the network together with its capability of keeping track of long-term dependencies. A similar approach has been proposed as Coupled Oscillatory Recurrent Neural Networks (coRNNs) (Rusch & Mishra, 2020) that are based on a secondary order ODEs modeling a coupled network of controlled forced and damped nonlinear oscillators. The authors prove precise bounds of the RNN's state gradients and thus the ability of the coRNN being a possible solution for exploding or vanishing gradients. Erichson et al. (2020) propose the Lipschitz RNN having additional hidden-to-hidden matrices enabling the RNN to remain Lipschitz continuous. This stabilizes the network and alleviates the exploding and vanishing gradient problem. In Li et al. (2020; 2021), the authors propose the Fourier respectively the Markov neural operator that are built from multiple concatenated Fourier layers that directly work on the Fourier modes of the dynamical system. This way, they retain major portion of the dynamics and forecast the future behavior of the system. Both, the incremental Recurrent Neural Network (IRNN) (Kag et al., 2019) and the time adaptive RNN (Kag & Saligrama, 2021) use additional recurrent iterations on each input to enable the model of coping different input time scales, where the latter provides a time-varying function that adapts the model's behavior to the time scale of the provided input.

All of this shows the increasing interest in the application of machine learning (ML) models for forecasting and analyzing (chaotic) dynamical systems. To meet this trend, Gilpin (2021) recently published a fully featured collection of benchmark datasets being related to chaotic systems including their mathematical properties.

A more general guide of training RNNs for chaotic systems is given by Monfared et al. (2021). They discuss under which conditions the chaotic behavior of the input destabilizes the RNN and thus leads to exploding gradients during training. As a solution they propose STF, where every $\tau$-th time step a true input value is provided (teacher forced) as input instead of the previous prediction.

## 4 Teaching Strategies

Within this section, we systematically discuss existing teaching strategies for sequence-to-sequence prediction models and propose new strategies. All of these will then be evaluated in an experimental study with different time series reported in the following section.

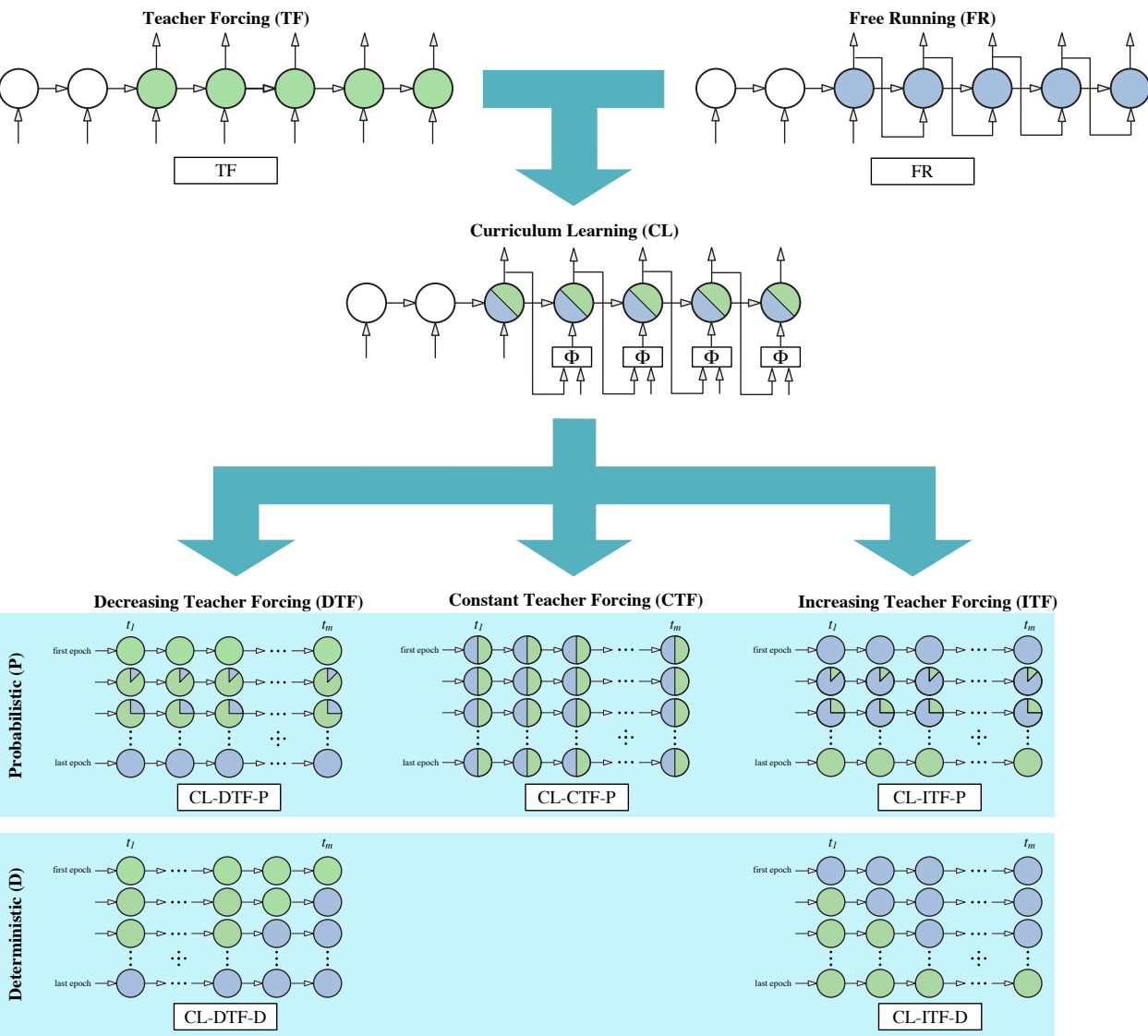

Figure 2: Overview of the proposed and evaluated training strategies where teacher forcing (TF) and free running (FR) refer to the two extreme cases that are combined to different CL strategies.

## 4.1 Free Running (FR) vs. Teacher Forcing (TF)

A widely used training strategy for RNN sequence-to-sequence models is to use TF throughout the whole training. Thereby, data is processed as shown in Fig. 2 (top left), i.e., the model is never exposed to its own predictions during training. A single forward step of the decoder during TF training is denoted as

$$\hat{y}^t = f(y^{t-1}, \theta, c^{t-1}), \tag{1}$$

where $y^t$ is the ground truth value for time step $t$, $\hat{y}^t$ is the predicted value for time step $t$, $\theta$ denotes the trainable parameters of the model, and $c^t$ is the decoder's hidden state at time step $t$.

The other extreme form of training is FR, i.e., only the model's first output value is predicted on basis of ground truth input and all subsequent output values of the sequence are predicted on basis of previous predictions throughout the training (cp. Fig. 2 (top right)). A single forward step of the decoder during FR training is denoted as

$$\hat{y}^t = f(\hat{y}^{t-1}, \theta, c^{t-1}). \tag{2}$$

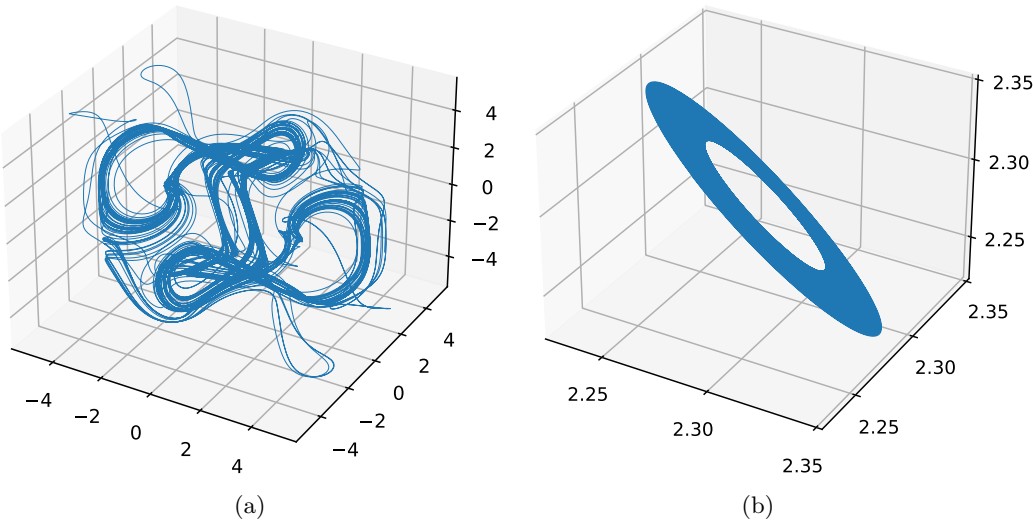

(a)                                                                          (b)

Figure 3: 30 000 time steps sampled with a time delta of $dt = 0.1$ of Thomas' cyclically symmetric attractor in (a) a chaotic parametrization and (b) a periodic parametrization

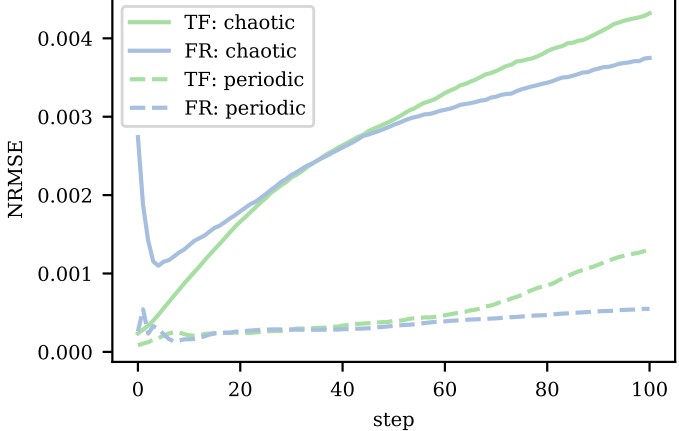

Figure 4: Test NRMSE over 100 predicted time steps of the chaotically and periodically parametrized Thomas attractor (cp. Fig. 3b), predicted by GRU models trained with teacher forcing (TF) and free running (FR).

A claimed major benefit of TF training is faster model convergence and thus reduced training time (Miao et al., 2020), while a major benefit of FR training is avoided exposure bias arising from solely training with ground truth data, yielding a model that performs less robust on unseen validation data (Ranzato et al., 2015). To illustrate these benefits and drawbacks, we utilize the Thomas attractor with two parametrizations, the first resulting in a periodic (cp. Fig. 3b) and the second resulting in a chaotic attractor (cp. Fig. 3a). By sampling from the attractors, we build two corresponding datasets of 10 000 samples each. For both datasets, we train a single layer encoder-decoder GRU following the FR and the TF strategy. Figure 4 shows the Normalized Root Mean Squared Error (NRMSE) per trained model over 100 predicted time steps. All models have been initialized with 150 ground truth values to build up the hidden state before predicting these 100 time steps. We observe that the chaotic variant is harder to predict for the trained models (cp. blue and green line in the figure) and that those trained with TF tend to predict with a smaller error at the first steps, which then grows relatively fast. In contrast, the prediction error of the FR-trained models starts on a higher level but stays more stable over the prediction horizon, arguing that time series forecasting represents an especially challenging task for sequence-to-sequence models when it exhibits chaotic behavior.

The more precise forecasting capabilities of a TF-trained network at the early prediction steps vs. the overall more stable long-term prediction performance of a FR-trained network observed in the Thomas example (cp. Fig 3a, 3b), motivates the idea of combining both strategies into a CL approach.

Schmidt (2019) describes the exposure bias in natural language generation as a lack of generalization. This argumentation motivates an analysis of training with FR and TF strategies when applied to forecasting dynamical systems with different amounts of available training data. Figure 5 shows the NRMSE when forecasting the Thomas attractor using different dataset sizes and reveals that increasing the dataset size yields generally improved model performance for TF as well as FR, while their relative difference is maintained.

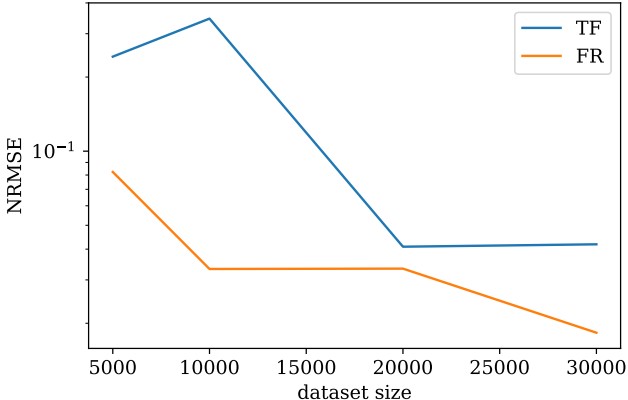

Figure 5: The NRMSE for different dataset sizes when using TF and FR during training for forecasting the Thomas attractor

## 4.2 Curriculum Learning (CL)

Within the context of our work, we denote the curriculum learning concept as combining TF and FR training, i.e., starting from the second decoder step, the curriculum prescribes per decoder step whether to use the ground truth value or the predicted value of the previous time step as input. We formalize a single training step of a CL approach as follows:

$$\hat{y}^t = \begin{cases} f(y^{t-1}, \theta, c^{t-1}), & \text{if } \Phi = 1 \\ f(\hat{y}^{t-1}, \theta, c^{t-1}), & \text{otherwise} \end{cases} \tag{3}$$

where the TF decision $\Phi$ governs whether the decoder input is teacher forced or not. Figure 2 illustrates the data flow of a sequence-to-sequence model training with CL in between the conventional strategies. In our naming scheme CL-DTF-P resembles the scheduled sampling approach proposed by Bengio et al. (2015). Below, we discuss the different types of curricula on training and iteration scale resulting in different ways for determining $\Phi$.

## 4.3 Curriculum on Training Scale

The TF ratio $\epsilon_i$ per training iteration $i$ is determined by a curriculum function $C : \mathbb{N} \rightarrow [0, 1]$. We distinguish three fundamental types of curriculum on training scale. First, constant curricula, where a constant amount of TF is maintained throughout the training denoted as

$$\epsilon_i = \epsilon. \tag{4}$$

Second, decreasing curricula, where the training starts with a high amount of TF that continuously declines throughout the training. Third, increasing curricula, where the training starts at a low amount of TF that

continuously increases throughout the training. Both follow a transition function $C : \mathbb{N} \rightarrow [\epsilon_{start}, \epsilon_{end}]$ denoted as

$$\epsilon_i = C(i), \tag{5}$$

where $\epsilon_{start} \leq \epsilon_i \leq \epsilon_{i+1} \leq \epsilon_{end}$ for increasing curricula, $\epsilon_{start} \geq \epsilon_i \geq \epsilon_{i+1} \geq \epsilon_{end}$ for decreasing curricula and $\epsilon_{start} \neq \epsilon_{end}$ for both. The following equations exemplary specify decreasing curricula (cp. Eqs. 6–8) following differing transition functions inspired by those used to study the scheduled sampling approach (Bengio et al., 2015)

$$
\begin{aligned}
C_{lin}(i) &= \max(\epsilon_{end}, \epsilon_{end} + (\epsilon_{start} - \epsilon_{end}) \cdot (1 - \frac{i}{Ł})), \\
&\quad \text{with } \epsilon_{end} < \frac{Ł - 1}{Ł}, \quad 1 < Ł, \quad i \in \mathbb{N}, \tag{6} \\
C_{invSig}(i) &= \epsilon_{end} + (\epsilon_{start} - \epsilon_{end}) \cdot \frac{k}{k + e^{\frac{i}{k}}}, \\
&\quad \text{with } \epsilon_{end} < \epsilon_{start}, \quad 1 \leq k, \quad i \in \mathbb{N}, \tag{7} \\
C_{exp}(i) &= \epsilon_{end} + (\epsilon_{start} - \epsilon_{end}) \cdot k^i, \\
&\quad \text{with } \epsilon_{end} < \epsilon_{start}, \quad 0 < k < 1, \quad i \in \mathbb{N}, \tag{8}
\end{aligned}
$$

where the curriculum length parameter $Ł$ determines the pace as number of iterations which the curriculum $C_{lin}$ needs to transition from $\epsilon_{start}$ to $\epsilon_{end}$. The curricula $C_{invSig}$ and $C_{exp}$ have no such parameter since the functions never completely reach $\epsilon_{end}$ in theory. In practice though, we adapt the curriculum-specific parameter $k$ to stretch or compress these curricula along the iteration axis to achieve the same effect. Figure 6a exemplary visualizes three decreasing and three increasing curricula, following differing transition functions $C$ and being parametrized with $\epsilon_{start} = 1$ and $\epsilon_{end} = 0$ and $\epsilon_{start} = 0$ and $\epsilon_{end} = 1$ respectively. Furthermore, each is parametrized to have a curriculum length of $Ł = 1\,000$. Figure 6b shows examples of decreasing and increasing $C_{lin}$ with different $Ł$.

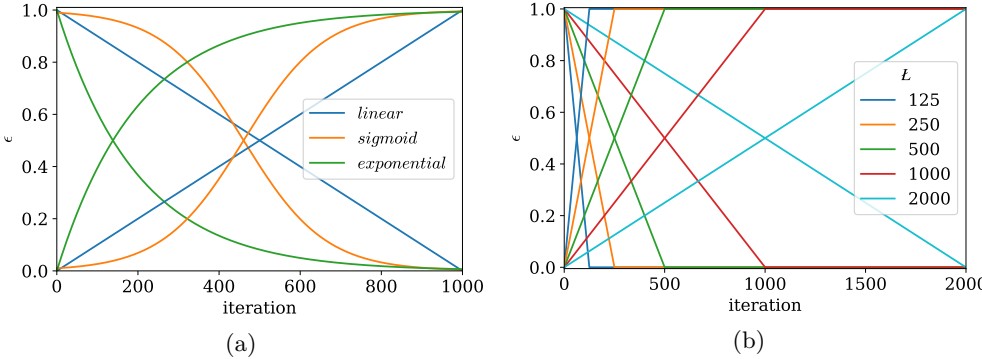

Figure 6: Examples of different decreasing curricula and their corresponding increasing versions (a) and multiple linear curricula with different pace $Ł$ (b).

## 4.4 Curriculum on Iteration Scale

$\epsilon_i$ prescribes a ratio of TF vs. FR steps for a given training iteration $i$. Based on $\epsilon$ that solely prescribes the amount of TF for an iteration, we can now develop micro curricula for distributing the TF and FR steps, eventually providing a TF decision $\Phi$ per training step. We propose two ways to distribute TF and FR steps within one training iteration: (1) probabilistic – where $\epsilon$ is interpreted as the probability of being a TF step, and (2) deterministic – where $\epsilon$ as a rate that determines the number of TF steps trained before moving to FR for the rest of the training sequence. For a probabilistic CL, we denote the TF decision $\Phi_\epsilon$, which is a discrete random variable that is drawn from a Bernoulli distribution:

$$\Phi_\epsilon \sim \text{Bernoulli}(\epsilon). \tag{9}$$

For a deterministic CL, $\Phi$ depends not only on $\epsilon$ but also on the current position $j$ within the predicted sequence of length $m$. Therefore, in this case we denote the TF decision $\Phi_{\epsilon,j}$ as:

$$\Phi_{\epsilon,j} = \begin{cases} 1, & \text{if } \epsilon \geq \frac{j}{m} \\ 0, & \text{otherwise.} \end{cases} \tag{10}$$

## 5 Evaluation

To compare the training strategies described in Section 4, we evaluate each with varying parametrization on six different datasets. Our experiments aim to answer the following six research questions:

**RQ1 Baseline teaching strategies.** How well and consistent do the current baseline strategies FR and TF train a model for forecasting dynamical systems?

**RQ2 Curriculum learning strategies.** How do the different curriculum learning strategies perform in comparison to the baseline strategies?

**RQ3 Training length.** How is training length influenced by the different teaching strategies?

**RQ4 Prediction stability.** How stable is a model's prediction performance over longer prediction horizons when trained with the different strategies?

**RQ5 Curriculum parametrization.** How much does the curriculum's parametrization influence model performance?

**RQ6 Iteration scale curriculum.** How do iteration scale curricula differ in resulting model performance?

### 5.1 Evaluated Curricula

In total, we define eight strategies to evaluate in this study (cp. Fig. 2). For comparison, we train the two baseline methods TF and FR that "teach" throughout the entire training or do not "teach" at all respectively. All other methods prescribe a teaching curriculum CL throughout the training and we distinguish these strategies along two dimensions: (1) the overall increasing (ITF), constant (CTF), or decreasing (DTF) trend in TF throughout the training curriculum and (2) the probabilistic (P) or deterministic (D) TF distribution within training steps.

### 5.2 Parametrization of Training Curricula

Table I shows all training-strategy-specific parameters and their values for the evaluated strategies. We subdivide our experiments into three sets: *baseline*, *exploratory*, and *essential* experiments. The baseline strategies FR and TF do not have any additional parameters. The CL-CTF-P strategy has the $\epsilon$ parameter configuring the strategy's TF ratio. The increasing and decreasing strategies CL-DTF-x and CL-ITF-x are configured by $\epsilon_{start}$ and $\epsilon_{end}$ referring to the initial and eventual amount of TF and the function $C$ transitioning between both. Additionally, Ł determines the number of training epochs in between $\epsilon_{start}$ and $\epsilon_{end}$. For the *exploratory* experiments, we utilize a fix Ł $= 1\,000$, while for the *essential* experiments, we evaluate all strategies solely using a linear transition $C_{linear}$ in the curriculum (cp. Eq. 6) with either $\epsilon_{start} = 0$ and $\epsilon_{end} = 1$ (increasing) or $\epsilon_{start} = 1$ and $\epsilon_{end} = 0$ (decreasing).

### 5.3 Performance Metrics

We use the NRMSE and the $R^2$ metrics as well as two derived of those to evaluate model performance. NRMSE is a normalized version of the Root Mean Squared Error (RMSE) where smaller values indicate better prediction performance. For a single value of a sequence, NRMSE is calculated as:

$$\text{NRMSE}(y, \hat{y}) = \frac{\sqrt{\frac{1}{d} \cdot \sum_{j=1}^{d}(y_j - \hat{y}_j)^2}}{\sigma}, \tag{11}$$

Table I: Curriculum strategy parameters used during the *baseline*, *exploratory*, and the *essential* experiments

|  | Strategy | parameter | Curriculum evaluated values |
|---|---|---|---|
| *baseline* | FR | C
$\epsilon$
Ł | –
0.0
– |
|  | TF | C
$\epsilon$
Ł | –
1.0
– |
| *exploratory* | CL-CTF-P | C
$\epsilon$
Ł | –
$\{0.25, 0.5, 0.75\}$
– |
|  | CL-DTF-P, CL-DTF-D | C
$\epsilon_{start} \to \epsilon_{end}$
Ł | $\{$linear, inverse sigmoid, exponential$\}$
$\{0.25, 0.5, 0.75, 1.0\} \to\ 0.0$
1000 |
|  | CL-ITF-P, CL-ITF-D | C
$\epsilon_{start} \to \epsilon_{end}$
Ł | $\{$linear, inverse sigmoid, exponential$\}$
$\{0.0, 0.25, 0.5, 0.75\} \to\ 1.0$
1000 |
| *essential* | CL-DTF-P, CL-DTF-D,
CL-ITF-P, CL-ITF-D | C
$\epsilon_{start} \to \epsilon_{end}$
Ł | linear
$\{0.0 \to 1.0, 1.0 \to 0.0\}$
$\{62, 125, 250, 500, 1\,000, 2\,000,$
$4\,000, 8\,000, 16\,000, 32\,000\}$ |

where $y$ is a ground truth vector, $\hat{y}$ is the corresponding prediction, $\sigma$ is the standard deviation across the whole dataset, and $d$ is the size of the vectors $y$ and $\hat{y}$. For model evaluation, we calculate the mean NRMSE over all $m$ forecasted steps of a sequence. Additionally, we compute and report the NRMSE only for the last $\lceil \frac{m}{10} \rceil$ forecasted steps of a sequence to specifically evaluate model performance at long prediction horizons.

The $R^2$ score lies in the range $(-\infty, 1]$ with higher values referring to better prediction performance. A score of 0 means that the prediction is as good as predicting the ground truth sequence's mean vector $\bar{y}$. The $R^2$ score is computed as:

$$R^2 = 1 - \frac{\sum_{j=1}^{d}(y_j - \hat{y}_j)^2}{\sum_{j=1}^{d}(y_j - \bar{y}_j)^2}. \tag{12}$$

We use the $R^2$ score to compute another metric $LT R^2 > 0.9$ measuring the number of Lyapunov Time (LT)s that a model can predict without the $R^2$ score dropping below a certain a threshold of 0.9. Sangiorgio and Dercole (Sangiorgio & Dercole, 2020) proposed this metric while applying a less strict threshold of 0.7.

### 5.4 Evaluated Datasets

We use six different time series datasets that we built by approximating six commonly studied chaotic systems (cp. Tab. II), i.e., Mackey-Glass (Mackey & Glass, 1977), Rössler (Rössler, 1976), Thomas' cyclically symmetric attractor (Thomas, 1999), Hyper Rössler (Rossler, 1979), Lorenz (Lorenz, 1963) and Lorenz'96 (Lorenz, 1996). Table II shows the differential equations per system and how we parametrized them. These systems differ from each other in the number of dimensions $d$ and the degree of chaos as indicated by the largest lyapunov exponent in the LLE column of Tab. II. The LLEs are approximated values that were published independently in the past (Brown et al., 1991; Sprott & Chlouverakis, 2007; Sano & Sawada, 1985; Sandri, 1996; Hartl, 2003; Brajard et al., 2020). We generate datasets by choosing an initial state vector of size $d$ and approximate $10\,000$ samples using the respective differential equations. We use SciPy package's implementation of the Livermore solver for ordinary differential equations (LSODE) (Radhakrishnan &

Table II: Details of the chaotic systems that were approximated to generate the data used for our experiments

| System | ODE/DDE | Parameters | $d$ | LLE |
|---|---|---|---|---|
| Mackey-Glass | $\frac{dx}{dt} = \beta \frac{x_\tau}{1+x_\tau^n} - \gamma x$ with $\gamma, \beta, n > 0$ | $\tau = 17$, $n = 10$, $\gamma = 0.1$ $\beta = 0.2$, $dt = 1.0$ | 1 | 0.006 |
| Thomas | $\frac{dx}{dt} = sin(y) - bx$ $\frac{dy}{dt} = sin(z) - by$ $\frac{dz}{dt} = sin(x) - bz$ | $b = 0.1$, $dt = 0.1$ | 3 | 0.055 |
| Rössler | $\frac{dx}{dt} = -(y + z)$ $\frac{dy}{dt} = x + ay$ $\frac{dz}{dt} = b + z(x - c)$ | $a = 0.2$, $b = 0.2$ $c = 5.7$, $dt = 0.12$ | 3 | 0.069 |
| Hyper Rössler | $\frac{dx}{dt} = -y - z$ $\frac{dy}{dt} = x + ay + w$ $\frac{dz}{dt} = b + xz$ $\frac{dw}{dt} = -cz + dw$ | $a = 0.25$, $b = 3$ $c = 0.5$, $d = 0.05$ $dt = 0.1$ | 4 | 0.14 |
| Lorenz | $\frac{dx}{dt} = -\sigma x + \sigma y$ $\frac{dy}{dt} = -xz + \rho x - y$ $\frac{dz}{dt} = xy - \beta z$ | $\sigma = 10$, $\beta = \frac{8}{3}$ $\rho = 28$, $dt = 0.01$ | 3 | 0.905 |
| Lorenz'96 | $\frac{dx_k}{dt} = -x_{k-2}x_{k-1} + x_{k-1}x_{k+1} - x_k + F$ for $k = 1 \ldots d$ and $x_{-1} = x_d$ | $F = 8$, $dt = 0.05$ | 40 | 1.67 |

Hindmarsh, 1993) except for Mackey-Glass which we approximate through the Python module JiTCDDE implementing the delayed differential equation (DDE) integration method, as proposed by (Shampine & Thompson, 2001). Thereby, $dt$ defines the time difference between two sampled states per dataset and is shown in Table II. Where available, we chose $dt$ similar to previous studies aiming for comparability of results. We split each dataset into 80% training samples and 10% validation and testing samples respectively. All data is normalized following a $z$-transform.

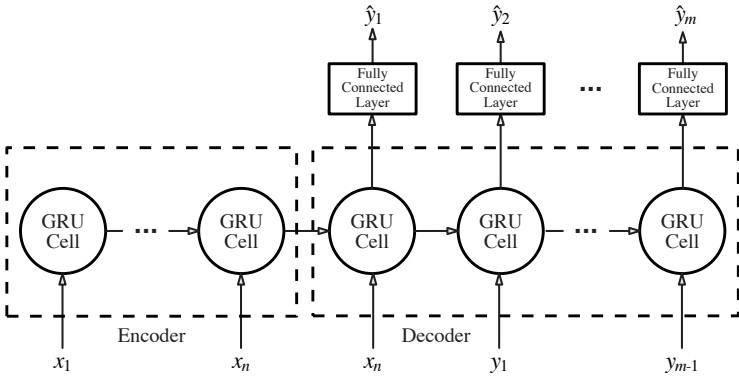

Figure 7: Structure of a simple encoder-decoder GRU used for training with teacher forcing

## 5.5 Training Procedure

All evaluated models follow an encoder-decoder GRU architecture with an additional fully connected layer after the decoder (cp. Fig. 7). We performed a full grid search for the hyper-parameters learning rate, batch size, learning rate reduction factor, loss plateau, input length $n$ and hidden state size to determine suitable configurations for the experiments. Based on this optimization, we used the Adam (Kingma et al., 2015) optimizer with a batch size of 128 and apply Reduce Learning Rate on Plateau (RLROP) with an initial

Table III: Results of the *exploratory* tests with the best hyper-parameter configuration per strategy and system. The arrow besides each metric's column title indicates whether smaller (↓) or larger (↑) values are favored. The best result values per dataset are printed in bold and the best baseline NRMSEs are underlined. Together with each dataset, we put the corresponding LLE in parenthesis.

| | Strategy | Best performing curriculum $C$ | $\epsilon$ | Trained epochs | NRMSE over 1LT absolut ↓ | rel. impr. ↑ | @BL epoch ↓ | last 10% ↓ |
|---|---|---|---|---|---|---|---|---|
| Thomas (0.055) | FR | constant | 0.00 | 427 | 0.03416 | – | – | 0.047222 |
| | TF | constant | 1.00 | 163 | 0.34545 | – | – | 0.607954 |
| | CL-CTF-P | constant | 0.25 | 450 | 0.05535 | −62.03% | 0.05675 | 0.082443 |
| | CL-DTF-P | inverse sigmoid | 0.75 ↘ 0.00 | 598 | 0.01858 | 45.61% | 0.02120 | 0.034325 |
| | CL-DTF-D | exponential | 0.25 ↘ 0.00 | 557 | 0.03229 | 5.47% | 0.03792 | 0.039749 |
| | CL-ITF-P | exponential | 0.00 ↗ 1.00 | 620 | 0.01403 | 58.93% | **0.02026** | 0.026014 |
| | CL-ITF-D | exponential | 0.25 ↗ 1.00 | 944 | **0.01126** | **67.04%** | 0.02179 | **0.018571** |
| Rössler (0.069) | FR | constant | 0.00 | 3 863 | 0.00098 | – | – | 0.000930 |
| | TF | constant | 1.00 | 500 | 0.00743 | – | – | 0.016119 |
| | CL-CTF-P | constant | 0.25 | 2 081 | 0.00084 | 14.29% | 0.00084 | 0.001333 |
| | CL-DTF-P | linear | 1.00 ↘ 0.00 | 2 751 | 0.00083 | 15.31% | 0.00083 | 0.000931 |
| | CL-DTF-D | inverse sigmoid | 0.25 ↘ 0.00 | 4 113 | 0.00064 | 34.69% | 0.00066 | 0.000578 |
| | CL-ITF-P | inverse sigmoid | 0.00 ↗ 1.00 | 7 194 | 0.00025 | 74.49% | 0.00034 | **0.000358** |
| | CL-ITF-D | linear | 0.75 ↗ 1.00 | 5 132 | **0.00024** | **75.51%** | **0.00031** | 0.000390 |
| Lorenz (0.905) | FR | constant | 0.00 | 918 | 0.01209 | – | – | 0.013166 |
| | TF | constant | 1.00 | 467 | 0.00152 | – | – | 0.002244 |
| | CL-CTF-P | constant | 0.75 | 297 | 0.00167 | −9.87% | 0.00167 | 0.002599 |
| | CL-DTF-P | inverse sigmoid | 0.75 ↘ 0.00 | 522 | 0.00168 | −10.53% | 0.00162 | 0.002425 |
| | CL-DTF-D | inverse sigmoid | 1.00 ↘ 0.00 | 204 | 0.00187 | −23.03% | 0.00187 | 0.002823 |
| | CL-ITF-P | linear | 0.00 ↗ 1.00 | 750 | 0.00149 | 1.97% | 0.00217 | 0.002235 |
| | CL-ITF-D | inverse sigmoid | 0.75 ↗ 1.00 | 803 | **0.00124** | **18.42%** | **0.00132** | **0.002084** |
| Lorenz'96 (1.67) | FR | constant | 0.00 | 8 125 | 0.07273 | – | – | 0.126511 |
| | TF | constant | 1.00 | 4 175 | 0.03805 | – | – | 0.075583 |
| | CL-CTF-P | constant | 0.50 | 2 615 | 0.07995 | −110.12% | 0.07995 | 0.140700 |
| | CL-DTF-P | linear | 0.75 ↘ 0.00 | 939 | 0.04654 | −22.31% | 0.04654 | 0.087228 |
| | CL-DTF-D | linear | 0.75 ↘ 0.00 | 1 875 | 0.04381 | −15.14% | 0.04381 | 0.081025 |
| | CL-ITF-P | inverse sigmoid | 0.25 ↗ 1.00 | 4 787 | **0.01854** | **51.27%** | **0.02016** | **0.036651** |
| | CL-ITF-D | inverse sigmoid | 0.00 ↗ 1.00 | 3 263 | 0.02093 | 44.99% | 0.02196 | 0.040356 |

learning rate of $1e^{-3}$ and a reduction factor of 0.6, i.e., 40% learning rate reduction, given a loss plateau of 10 epochs for all datasets except Lorenz'96, where we use a reduction factor of 0.9 and a 20 epoch plateau respectively. Furthermore, we found an input length of $n = 150$ steps and a *hidden state size* of 256 to be most suitable. We use early stopping with a *patience* of 100 epochs and a *minimum improvement threshold* of 1% to ensure the convergence of the model while preventing from overfitting. We train all models with a dataset-specific prediction length $m$ defined as:

$$m = \left\lceil \frac{\mathrm{LT}}{dt} \right\rceil = \left\lceil \frac{1}{dt \cdot \mathrm{LLE}} \right\rceil. \tag{13}$$

The reason being that we aim to train for the same forecasting horizon that we mainly evaluate a trained model with. We adapt this horizon to the dataset's LT, thereby aiming for performance measures that are comparable across datasets.

We provide plots of the training and validation loss curves of the final parametrization per strategy and dataset in Appendix A. Based on these loss curves, we observe for ITF in contrast to DTF strategies that the training loss tends to move away from the validation loss faster. This is explainable by the fact that with increasing training time, ITF strategies deliver an increasing amount of TF inputs, counteracting the accumulation of error along the forecasted sequence and therefore further reducing training loss. For DTF strategies we observe an opposing behavior. Regarding training iterations, we observe that ITF strategies

typically train for a larger number of epochs. Following the TF ratio curve ($\epsilon$) we see that even when the increasing curriculum already arrived at the final $\epsilon_e = 1.0$, which means that it is using solely TF training from there on, it can improve for many more epochs than by simply using TF from the beginning. Since the termination of the training is determined by the early stopping criterion this shows that ITF may facilitate a longer and (regarding the validation loss) typically more successful training process compared to DTF and baseline strategies. We show the corresponding curves exemplary for the Rössler training in Figure 8.

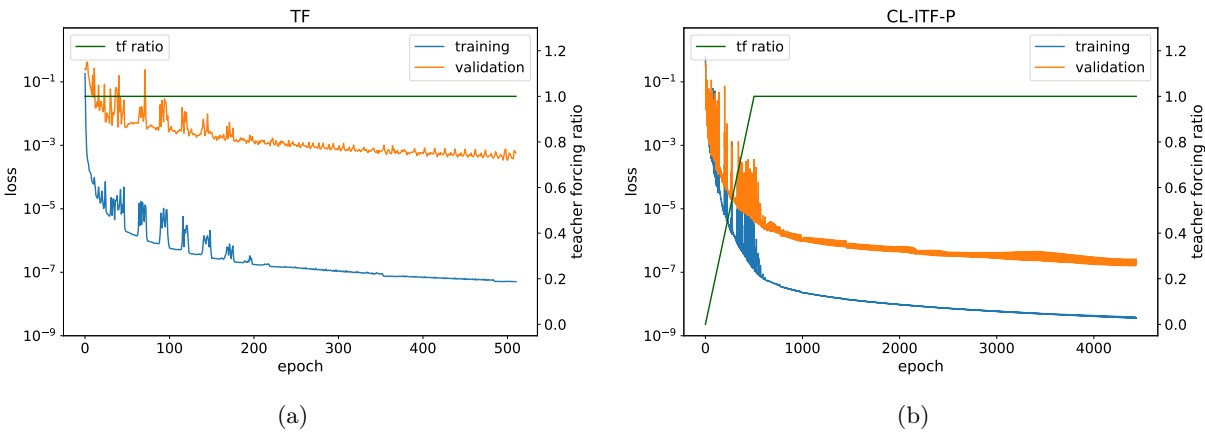

Figure 8: Loss and teacher forcing ratio curves for Rössler

## 5.6   Results

[1]Table III shows results for the *exploratory* experiments. Per evaluated strategy and dataset, the table reports resulting model performance in terms of NRMSE. We only report that curriculum configuration in terms of transition function $C$ and $\epsilon$ schedule that yields the highest NRMSE per strategy and dataset. Each model was used to predict a dataset-spetefic horizon of 1 LT. The best result per dataset and performance metric is highlighted in bold. First, we study the baseline strategies FR and TF and observe that for two datasets, i.e., Thomas and Rössler, the FR strategy outperforms the TF baseline, while TF outperforms FR for the other two. We select the one that performs best per dataset to measure relative improvement or deterioration gained by training with the respective curriculum learning strategy (cp. column "NMRSE rel. impr."). We observe that across all datasets and performance metrics, the CL-ITF-P and CL-ITF-D strategies yield the best and second best-performing model with a relative improvement of $1.97 - 80.61\%$ over the best performing baseline strategy. The other curriculum learning strategies perform less consistent across the datasets. The CL-DTF-x strategies yield an improved NRMSE for half of the datasets, while the constant CL-CTF-P only yields an improvement for the Thomas attractor. We separately report the NRMSE of the last 10% predicted values of 1 LT test horizon per dataset to assess how robust a prediction is over time (cp. column "NRMSE last 10%"). We observe that the CL-ITF-P and CL-ITF-D strategies also reach the best performance in terms of this metric, meaning that they yield the most robust models. We further observe a diverse set of curriculum configurations yielding the best performing model per strategy and dataset. That means that all available transition functions, i.e., linear, inverse sigmoid, and exponential, have been discovered as best choice for at least one of the trained models. Further, we observe all evaluated $\epsilon$ as best choice for the CL-CTF-P strategy and one of the datasets respectively. Similarly, the best-performing initial $\epsilon$ for the increasing and decreasing transitions per dataset spans all evaluated values except for 0.5. The table also reports the number of training iterations until reaching the early stopping criterion (cp. column "training epochs"). We observe that the two baseline strategies utilize strongly differing numbers of iterations across all datasets. For the Thomas and the Rössler attractor, the TF strategy does not allow for proper model convergence, being characterized by a low number of iterations and a high NRMSE compared to the other strategies.

---

[1]A reproduction package for the experiments is available on Github: https://github.com/phit3/flipped_classroom. The datasets used in this paper are published on Dataverse: https://doi.org/10.7910/DVN/YEIZDT.

Table IV: Best curriculum length per strategy and system for all six datasets. The arrow besides a metric's column title indicates whether smaller (↓) or larger (↑) values are favored. The best result values per dataset are printed in bold and the best baseline NRMSEs are underlined. Together with each dataset, we put the corresponding LLE in parenthesis.

| | Strategy | Best performing curriculum $\epsilon$ | Ł | Trained epochs | NRMSE over 1LT absolut↓ | rel. impr.↑ | @BL epoch↓ | last 10%↓ | #LT with $R^2 > 0.9$ ↑ |
|---|---|---|---|---|---|---|---|---|---|
| **Mackey Glass (0.006)** | FR | 0.00 | – | 4713 | _0.00391_ | – | 0.00391 | 0.004101 | 4.50 |
| | TF | 1.00 | – | 44 | 0.09535 | – | 0.09535 | 0.171945 | 1.64 |
| | CL-CTF-P | 0.25 | – | 2918 | 0.00632 | −61.64% | 0.00632 | 0.006544 | 4.51 |
| | CL-DTF-P | 1.00 ↘ 0.00 | 2000 | 3733 | 0.00215 | 45.01% | 0.00215 | 0.003010 | 4.95 |
| | CL-DTF-D | 1.00 ↘ 0.00 | 1000 | 431 | 0.00585 | −49.62% | 0.00585 | 0.011022 | 3.91 |
| | CL-ITF-P | 0.00 ↗ 1.00 | 500 | 1566 | **0.00104** | **73.40%** | **0.00104** | **0.001793** | **5.18** |
| | CL-ITF-D | 0.00 ↗ 1.00 | 500 | 1808 | 0.00211 | 46.03% | 0.00211 | 0.003032 | 4.99 |
| **Thomas (0.055)** | FR | 0.00 | – | 427 | _0.03416_ | – | 0.03416 | 0.047222 | 2.04 |
| | TF | 1.00 | – | 163 | 0.34545 | – | 0.34545 | 0.607954 | 1.73 |
| | CL-CTF-P | 0.25 | – | 450 | 0.05535 | −62.03% | 0.05675 | 0.082443 | 1.53 |
| | CL-DTF-P | 1.00 ↘ 0.00 | 1000 | 356 | 0.05084 | −48.83% | 0.05084 | 0.105585 | 2.13 |
| | CL-DTF-D | 1.00 ↘ 0.00 | 1000 | 326 | 0.10712 | −213.58% | 0.10712 | 0.206923 | 1.53 |
| | CL-ITF-P | 0.00 ↗ 1.00 | 500 | 677 | **0.00930** | **72.78%** | **0.01645** | **0.016729** | **3.99** |
| | CL-ITF-D | 0.00 ↗ 1.00 | 500 | 649 | 0.01819 | 46.75% | 0.03934 | 0.030589 | 2.05 |
| **Rössler (0.069)** | FR | 0.00 | – | 3863 | _0.00098_ | – | 0.00098 | 0.000930 | 9.46 |
| | TF | 1.00 | – | 500 | 0.00743 | – | 0.00743 | 0.016119 | 4.75 |
| | CL-CTF-P | 0.25 | – | 2081 | 0.00084 | 14.29% | 0.00084 | 0.001333 | 7.51 |
| | CL-DTF-P | 1.00 ↘ 0.00 | 1000 | 2751 | 0.00083 | 15.31% | 0.00083 | 0.000931 | 8.46 |
| | CL-DTF-D | 1.00 ↘ 0.00 | 125 | 4879 | 0.00100 | −2.04% | 0.00116 | 0.000947 | 9.28 |
| | CL-ITF-P | 0.00 ↗ 1.00 | 500 | 4523 | **0.00019** | **80.61%** | **0.00022** | **0.000303** | **10.23** |
| | CL-ITF-D | 0.00 ↗ 1.00 | 4000 | 7267 | 0.00027 | 72.24% | 0.00051 | 0.000368 | 9.41 |
| **Hyper Rössler (0.14)** | FR | 1.00 | – | 6461 | 0.00599 | – | 0.00599 | 0.007011 | 6.57 |
| | TF | 0.00 | – | 2788 | _0.00435_ | – | 0.00762 | 0.011194 | 5.24 |
| | CL-CTF-P | 0.25 | – | 2909 | 0.01450 | −233.33% | 0.01450 | 0.015944 | 5.21 |
| | CL-DTF-P | 1.00 ↘ 0.00 | 2000 | 3773 | 0.00560 | 28.74% | 0.00560 | 0.007052 | 6.32 |
| | CL-DTF-D | 1.00 ↘ 0.00 | 16000 | 1793 | 0.00490 | 12.64% | 0.00490 | 0.007471 | 6.30 |
| | CL-ITF-P | 0.00 ↗ 1.00 | 125 | 2802 | 0.00366 | 15.86% | 0.00366 | 0.005802 | 6.50 |
| | CL-ITF-D | 0.00 ↗ 1.00 | 250 | 3317 | **0.00326** | **25.06%** | **0.00326** | **0.004639** | **6.72** |
| **Lorenz (0.905)** | FR | 0.00 | – | 918 | 0.01209 | – | 0.01319 | 0.013166 | 3.31 |
| | TF | 1.00 | – | 467 | _0.00152_ | – | 0.00152 | 0.002244 | 6.72 |
| | CL-CTF-P | 0.75 | – | 297 | 0.00167 | −9.87% | 0.00167 | 0.002599 | 6.43 |
| | CL-DTF-P | 1.00 ↘ 0.00 | 4000 | 450 | 0.00124 | 18.42% | **0.00124** | 0.001925 | 6.64 |
| | CL-DTF-D | 1.00 ↘ 0.00 | 16000 | 587 | 0.00111 | 26.97% | 0.00127 | 0.001650 | 6.53 |
| | CL-ITF-P | 0.00 ↗ 1.00 | 250 | 1137 | **0.00060** | **60.53%** | **0.00124** | **0.000883** | **7.19** |
| | CL-ITF-D | 0.00 ↗ 1.00 | 250 | 578 | 0.00135 | 11.18% | 0.00189 | 0.001725 | 4.33 |
| **Lorenz'96 (1.67)** | FR | 0.00 | – | 8125 | 0.07273 | – | 0.08362 | 0.126511 | 2.34 |
| | TF | 1.00 | – | 4175 | _0.03805_ | – | 0.03805 | 0.075583 | 3.01 |
| | CL-CTF-P | 0.50 | – | 2615 | 0.07995 | −110.12% | 0.07995 | 0.140700 | 2.25 |
| | CL-DTF-P | 1.00 ↘ 0.00 | 1000 | 983 | 0.05278 | −38.71% | 0.05278 | 0.098130 | 2.67 |
| | CL-DTF-D | 1.00 ↘ 0.00 | 1000 | 4083 | 0.07119 | −87.10% | 0.07119 | 0.126636 | 2.34 |
| | CL-ITF-P | 0.00 ↗ 1.00 | 250 | 3886 | 0.01680 | 55.85% | 0.01680 | 0.032439 | 4.01 |
| | CL-ITF-D | 0.00 ↗ 1.00 | 250 | 3379 | **0.01628** | **57.21%** | **0.01628** | **0.031464** | **4.18** |

Among the curriculum teaching strategies across all datasets, the strategies with increasing TF ratio CL-ITF-x utilize the most training iterations. These CL-ITF-x strategies also utilize more training iterations than the better-performing baseline strategy across all datasets. To better understand whether the longer training is the sole reason for the higher performance of the CL-ITF-x trained models, we additionally report the performance in terms of NRMSE of all curriculum-trained models after the same number of training iterations as the better performing baseline model, i.e., after 427 epochs for Thomas, after 3863 epochs for Rössler, after 467 epochs for Lorenz, and after 4175 epochs for Lorenz'96 (cp. column "Performance @BL epochs"). We observe across all datasets that the best-performing teaching strategy still remains CL-ITF-P or CL-ITF-D. In conclusion, the *exploratory* experiments demonstrated that a well-parametrized CL-ITF-x strategy yields a 18.42 − 75.51% performance increase across the evaluated datasets.

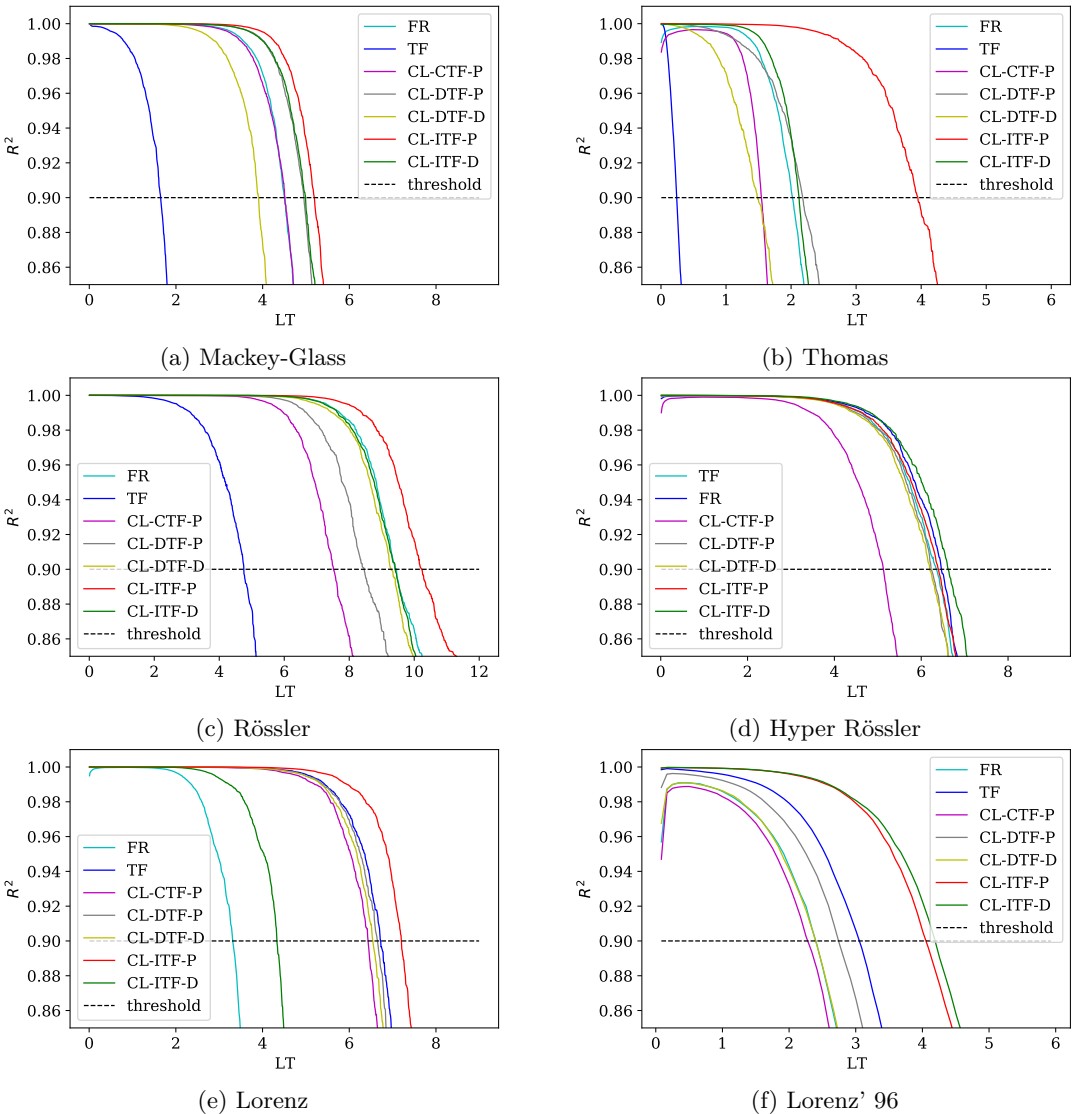

Figure 9: $R^2$ score over multiple LTs for the six studied datasets using eight different training strategies

However, this improvement comes at the cost of an intensive parameter optimization of the respective curriculum. Therefore, we run a second series of *essential* experiments in which we simplify the parametrization of the curriculum by utilizing a linear transition from either $0.0 \rightarrow 1.0$ (CL-ITF-x) or $1.0 \rightarrow 0.0$ (CL-DTF-x), that is solely parametrized by the length of this transition in terms of training epochs Ł. Table IV reports results in terms of the previously introduced performance metrics again, measured over a prediction horizon of 1 LT and across the same teaching strategies for six datasets, including those studied for the *exploratory* experiments. Since the changes of the *essential* over the *exploratory* experiments solely affect teaching strategies with a training-iteration-dependent curriculum, they have no effect on the baseline strategies FR and TF as well as the constant curriculum CL-CTF-P, which we still report in Table IV for direct comparison. Overall, we observe that CL-ITF-P outperforms all other strategies for four out of six datasets, while it performs second best for the remaining two datasets where the deterministic version CL-ITF-D performs best. These strategies yield relative improvements ranging from $25.06 - 80.61\%$ and are, thus, even higher than those observed for the *exploratory* experiments. Beyond that we observe that for all datasets treated in both experimental sets, the training curricula used in the *essential* experiments yield better performing models. For three out of four datasets, the training even requires substantially less training iterations than

in the explorative experiments. Additionally, we report in column "#LT with $R^2 > 0.9$" the prediction horizon in terms of LT that a trained model can predict while constantly maintaining an $R^2 > 0.9$. We observe that the ranking between the different strategies mostly remains the same as those observed when predicting 1 LT. That means that the best-performing strategy consistently remains the same for the longer horizon. Figure 9 more concretely depicts how the $R^2$ score develops across the prediction horizon for the different teaching strategies and datasets.

### 5.7 Additional Experiments

Judging from the essential experiments, CL-ITF-x are our winning strategies on the datasets we tested. However, as mentioned in Section 3, there are many other approaches that predict chaotic systems with adapted RNNs architectures that take theoretical insights of dynamical systems into account. STF (Monfared et al., 2021) does not require any architectural modifications but instead provides an adapted training strategy. It determines a time interval $\tau = \frac{ln2}{LLE}$ that denotes how many FR steps are processed before the next TF value is used within one sequence. We find this strategy comparable to the CL approaches we study. Therefore, we execute another set of experiments where we use STF during the training of our encoder-decoder GRU for all chaotic systems in Table II. Since our data is sampled with different $dt$, we have to redefine the time interval as $\tau = \frac{ln2}{LLE \cdot dt}$.

The results (cp. Tab. V) show that STF provides improved performance compared to the best baseline for three of six datasets ranging from $26.21 - 46.75\%$ relative improvement. For this it requires no additional hyper-parameters if the system's LLE is known. It also beats the best-performing CL strategy on the Hyper-Rössler dataset by a margin of 1.15%. For the rest of the datasets, the results stay behind those of the CL-ITF-x strategies, showing a worse, i.e., increased, NRMSE by $7.00 - 236.18\%$. We assume that where STF systematically induces TF to catch chaos-preventing exploding gradients before they appear, using knowledge about the processed data, CL helps the model to find more consistent minima while not taking the degree of chaos into account. Originally, STF was not proposed to work for the kind of sequence-2-sequence architecture we are using, which may limit its effectiveness here. We further hypothesize that the GRU is able to keep the risk of exploding gradients low in many cases, due to its gating mechanism and thus prevents STF to really show its full strength here.

For further investigation on CL for non-chaotic systems and to enrich our experiments, we conduct additional experiments that include the application of the baseline strategies TF and FR together with CL-ITF-P and CL-ITF-D on a periodic system and a measured real-world dataset. We use CL-ITF-P and CL-ITF-D, since they provide the most consistent relative improvements in the essential experiments. As periodic system, we study the Thomas attractor (Thomas, 1999) with parameter $b = 0.32899$, which ensures a periodic behavior. Extending our evaluation to empirical data, we selected a time series used in the Santa Fe Institute competition (Weigend & Gershenfeld, 1993)[2].

The results in Table VI support our assumption that CL-ITF-x strategies are also successfully applicable for time series data originating dynamical systems with periodic behavior, achieving relative improvements of $21.05 - 42.11\%$. Regarding the Santa Fe dataset we observe less impact by our strategies. Only having an improvement by 5.90% for CL-ITF-P and a worsening by 2.86% for CL-ITF-D on the empirical real-world data. When increasing the prediction horizon from 20 to 200 steps, we observe an increased relative improvement of 26.27% for CL-ITF-P. At the same time, we observe a relative decrease of $-5.89\%$ for CL-ITF-D. To investigate the radical difference between these results, we study the $\epsilon$ in relation to the appearance of TF steps in the respective training curriculum. Figure 10 shows two different probabilities along the training epochs. There is the TF probability $\epsilon$ that is directly determined by the curriculum function and $p_{\leq 200}$, which denotes the probability of a training sample to contain at least one TF step. It is computed as $p_{\leq 200} = \epsilon \sum_{i=1}^{200}(1-\epsilon)^{i-1}$ for the probabilistic strategy. For CL-ITF-D, this probability is either 0 or 1, meaning that it only has an effect on training epochs $\geq 160$. If we would use the same Ł as for CL-ITF-P here, the training would converge after 208 epochs just as in the plain FR case. CL-ITF-P though, due to its probabilistic nature, can sporadically induce TF steps for some of the batches in the

---

[2]https://github.com/tailhq/DynaML/blob/master/data/santafelaser.csv

Table V: Results of STF with those of the baseline and the CL-ITF-x strategies

| System | Strategy | Epochs | NRMSE ↓ | Rel. impr. ↑ |
|---|---|---|---|---|
| Mackey-Glass (0.006) | FR | 4 713 | 0.00391 | – |
| | TF | 44 | 0.09535 | – |
| | CL-ITF-P | 1 566 | 0.00104 | 73.40% |
| | CL-ITF-D | 1 808 | **0.00211** | 46.03% |
| | STF | 5 517 | 0.00254 | 35.04% |
| Thomas (0.055) | FR | 427 | 0.03416 | – |
| | TF | 163 | 0.34545 | – |
| | CL-ITF-P | 677 | **0.00930** | 72.78% |
| | CL-ITF-D | 649 | 0.01819 | 46.75% |
| | STF | 432 | 0.03655 | −7.00% |
| Rössler (0.069) | FR | 3 863 | 0.00098 | – |
| | TF | 500 | 0.00743 | – |
| | CL-ITF-P | 4 523 | **0.00019** | 80.61% |
| | CL-ITF-D | 7 267 | 0.00027 | 72.24% |
| | STF | 4 796 | 0.00065 | 33.67% |
| Hyper-Rössler (0.14) | FR | 6 461 | 0.00599 | – |
| | TF | 2 788 | 0.00435 | – |
| | CL-ITF-P | 2 802 | 0.00366 | 15.86% |
| | CL-ITF-D | 3 317 | 0.00326 | 25.06% |
| | STF | 2 645 | **0.00321** | 26.21% |
| Lorenz (0.905) | FR | 9 18 | 0.01209 | – |
| | TF | 4 67 | 0.00152 | – |
| | CL-ITF-P | 1 137 | **0.00060** | 60.53% |
| | CL-ITF-D | 5 78 | 0.00135 | 11.18% |
| | STF | 1 853 | 0.00511 | −236.18% |
| Lorenz'96 (1.67) | FR | 8 125 | 0.07273 | – |
| | TF | 4 175 | 0.03805 | – |
| | CL-ITF-P | 3 886 | 0.01680 | 55.85% |
| | CL-ITF-D | 3 379 | **0.01628** | 57.21% |
| | STF | 1 478 | 0.09030 | −137.32% |

earlier phase of training and thus provides a much smoother transition towards TF phase. We argue that this smoothness is crucial for the given task, which appears to be very sensitive to TF training steps.

We also conducted experiments for other RNN architectures, i.e., we selected a vanilla RNN, an LSTM, a unitary evolution RNN (uRNN) (Arjovsky et al., 2016) and a Lipschitz RNN (LRNN) (Erichson et al., 2020), in the same encoder-decoder setup as applied for the previous experiments with the GRU architecture. In the Table VII, we compare the NRMSE and relative improvement of these architectures on the four chaotic datasets from the exploratory experiments (cp. Tab. III). We compare TF, FR and previously best-performing training strategies CL-ITF-P and CL-ITF-D (cp. Tab. IV). Except for one case, we observe that both CL strategies outperform the respective best-performing baseline strategy on the vanilla RNN as well as the LSTM architecture. The only exception is a vanilla RNN trained to forecast the Lorenz'96 system using CL-ITF-P. This setup suffers a performance decrease by 83.45%, while the NRMSE in all other experiments

Table VI: Comparing baseline strategies and CL-ITF-P on periodic Thomas and measured Santa Fe laser dataset

|  | Strategy | Ł | Epochs | NRMSE ↓ | Rel. impr. ↑ |
|---|---|---|---|---|---|
| Per. Thomas | FR | – | 542 | 0.00057 | – |
|  | TF | – | 542 | 0.00107 | – |
|  | CL-ITF-P | 8 000 | 794 | **0.00033** | 42.11% |
|  | CL-ITF-D | 125 | 326 | 0.00045 | 21.05% |
| Santa Fe ($m = 20$) | FR | – | 500 | 0.02170 | – |
|  | TF | – | 22 | 0.04793 | – |
|  | CL-ITF-P | 32 000 | 469 | **0.02042** | 5.90% |
|  | CL-ITF-D | 4 000 | 536 | 0.02232 | −2.86% |
| Santa Fe ($m = 200$) | FR | – | 208 | 0.2786 | – |
|  | TF | – | 1 | 0.7104 | – |
|  | CL-ITF-P | 64 000 | 1506 | **0.2054** | 26.27% |
|  | CL-ITF-D | 32 000 | 201 | 0.2950 | −5.89% |

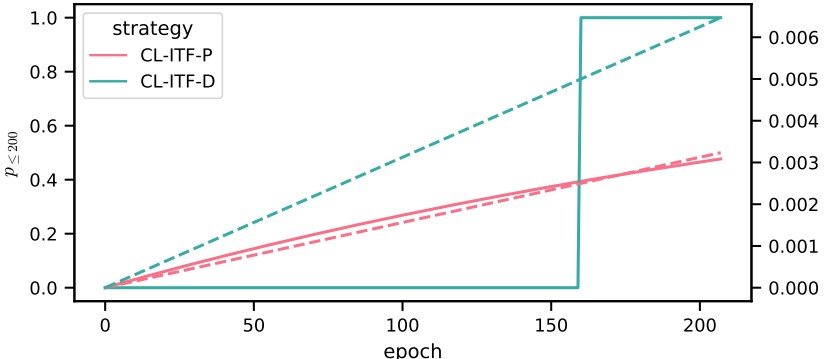

Figure 10: The probability course of a sample to include at least one TF step (solid) and the corresponding TF probability $\epsilon$ (dashed). Exemplary shown for the Santa Fe setup with 200 output steps (cp. Tab. VI).

improves by $37.83 – 75.28\%$ RNN and $26.04 – 69.75\%$ LSTM respectively when using the CL strategies. The uRNN and the LRNN do not adapt their weight matrices directly while training, but rather update several structured building block matrices to construct the actual weight matrices. We observe improvements from 0.4 to $77.75\%$ and from 0.21 to $83.74\%$ for the two architectures respectively. We provide more detailed information about these experiments in appendix C.

In our experiments on the Lorenz'96 data, the performance of the encoder-decoder RNN remains an outlier. Especially when we consider that for the deterministic variant CL-ITF-D there is a clear NRMSE improvement by $75.28\%$. We use this setup to take a closer look on the training gradients and process. In Figure 11a, we plot the mean of the decoder RNN gradient norm ($||\nabla W_d||$) over the training epochs for the CL-ITF-P and the CL-ITF-D case. We also add the TF probability $\epsilon$ and the learning rate $\eta$ to the plot. For comparison with the two successful setups, we also give an overview of the gradient norm of the GRU and the LSTM-based model under the same conditions (cp. Figs 11c and 11b).

For all models, we observe an oscillation of the gradient norm during CL-ITF-P training. CL-ITF-D avoids this behavior with a constant iteration scale curriculum that keeps the TF-FR-pattern fix for several epochs between the changes. The crucial epochs are indicated by green dashed vertical lines. In contrast, CL-ITF-P provides a probabilistic iteration scale curriculum that leads to intra-epoch changes. The resulting variation

in the gradient norm only stops when the training scale curriculum hits the final $\epsilon = 1$. Preceding that, the oscillating gradient norm originates from an unsteady training loss. Most of the time, the unsteady loss can be compensated due to the increasing amount of TF over the epochs. For the Lorenz'96 data, however, it seems the unsteady loss together with the used scheduler lead to a rapid decrease of the learning rate. This effect is observable during the curriculum training of the basic RNN, the LSTM and the GRU. The latter two models seem to handle the resulting minimal learning rate of $3e^{-6}$ much better and manage to continue their training successfully.

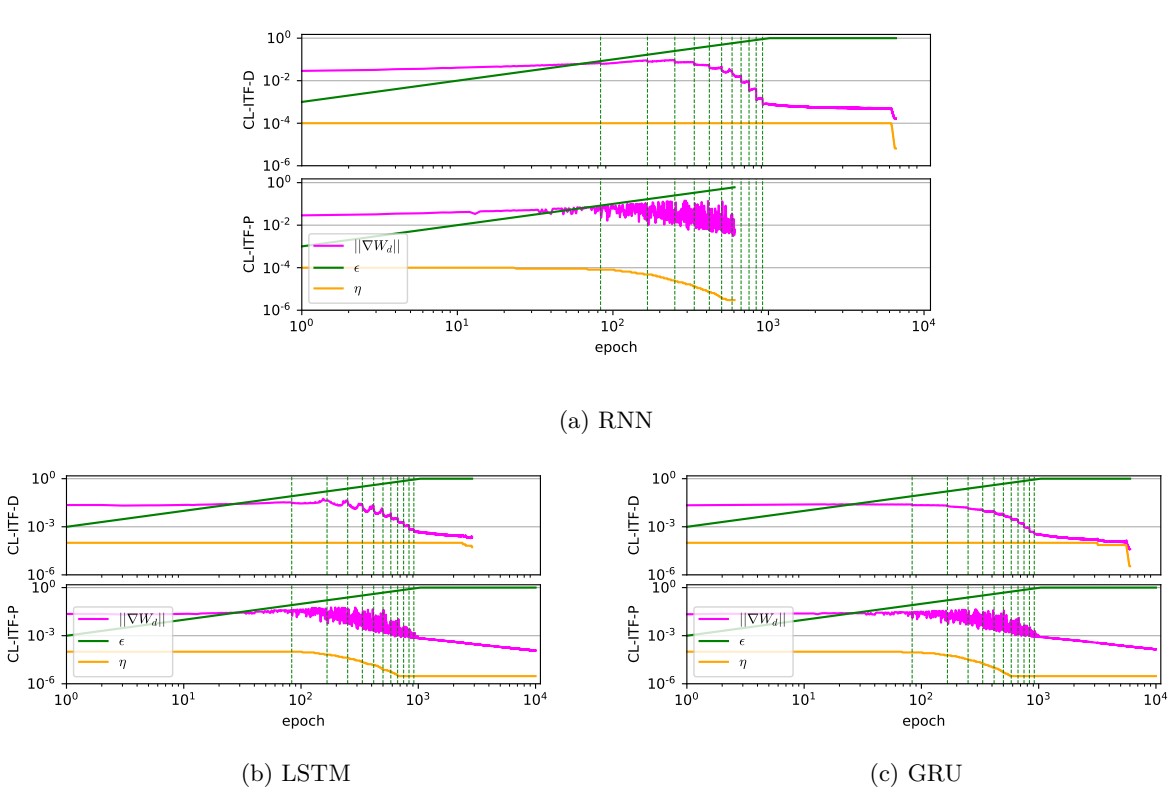

Figure 11: Course of the average decoder gradient norm $||\nabla W_d||$ during training on Lorenz'96 data together with TF probability $\epsilon$ and learning rate $\eta$.

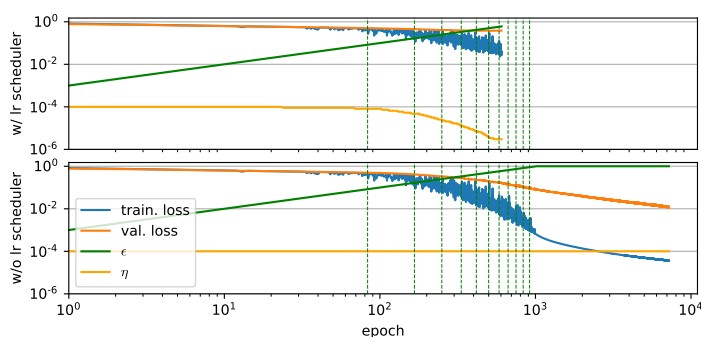

Figure 12: Course of the training and validation loss for the RNN on Lorenz'96 data using CL-ITF-P with and without learning rate scheduler.

To get an impression of the RNN's behavior without the unwanted learning rate drop, we conducted the same CL-ITF-P experiments again without applying any learning rate scheduler. This time we plot the training loss, that exhibits the same oscillating behavior as the gradient norm, together with the validation loss, to see if the change leads to improved loss values. The plot in Figure 12 shows that without the unwanted learning rate drop, the basic RNN remains stable during the transient phase towards $\epsilon = 1$ and ends up with a validation loss of 0.012 instead of 0.373. For the more sophisticated RNNs, we see clearly improved validation losses after much less epochs as well. The corresponding plots are shown in appendix B.

Table VII: Forecasting performance of additional RNNs (vanilla RNN, LSTM, unitary evolution RNN (uRNN) and Lipschitz RNN (LRNN)) on the different chaotic datasets. Given is the NRMSE with relative improvement in parenthesis for the CL strategies.

| System | Strategy | RNN | LSTM | uRNN | LRNN |
|---|---|---|---|---|---|
| Thomas | FR | 0.48117 | 0.43698 | 0.74991 | 0.11697 |
| | TF | 0.41274 | 0.05265 | 0.85679 | 0.05714 |
| | CL-ITF-P | **0.17955** (56.50%) | 0.01892 (64.06%) | **0.61889** (17.47%) | 0.05237 (8.35%) |
| | CL-ITF-D | 0.25659 (37.83%) | **0.01181** (77.57%) | 0.73331 (2.21%) | **0.03926** (31.29%) |
| Rössler | FR | 0.00747 | 0.00210 | 0.03726 | 0.27122 |
| | TF | 0.50174 | 0.00139 | 0.03317 | 0.90844 |
| | CL-ITF-P | **0.00283** (62.12%) | **0.00063** (54.68%) | 0.02473 (25.44%) | 0.05185 (80.88%) |
| | CL-ITF-D | 0.00319 (57.30%) | 0.00085 (38.85%) | **0.02420** (27.04%) | **0.04409** (83.74%) |
| Lorenz | FR | 0.11389 | 0.06526 | 0.13021 | 0.06675 |
| | TF | 0.00913 | 0.00169 | 0.96357 | 0.02830 |
| | CL-ITF-P | 0.00603 (33.95%) | **0.00075** (55.62%) | 0.03098 (76.21%) | **0.01732** (38.80%) |
| | CL-ITF-D | **0.00378** (58.60%) | 0.00125 (26.04%) | 0.02897 (77.75%) | 0.02417 (14.59%) |
| Lorenz'96 | FR | 0.31002 | 0.13010 | 0.26359 | 0.62710 |
| | TF | 0.57870 | 0.22757 | 0.61159 | 0.65122 |
| | CL-ITF-P | 0.56872 (−83.45%) | **0.03935** (69.75%) | 0.26253 (0.4%) | **0.62170** (0.86%) |
| | CL-ITF-D | **0.07663** (75.28%) | 0.06855 (47.31%) | **0.18483** (29.88%) | 0.62581 (0.21%) |

## 6 Discussion

**Baseline teaching strategies (RQ1).** Considering the baseline teaching strategies FR and TF, we observe that per dataset, one of the strategies performs substantially better than the other. We also observe that for the upper, based on their LLE, less chaotic datasets in Table IV FR performs better, while for the lower, more chaotic datasets, TF yields the better-performing model. However, a larger study with more datasets would be required to justify this claim. Our takeaway is that none of the methods can be universally recommended again motivating curriculum learning strategies.

**Curriculum learning strategies (RQ2).** Among the curriculum learning strategies, we observe that blending FR with a constant ratio of TF, i.e., CL-CTF-P, almost consistently yields worse results than

the best-performing baseline strategy and we therefore consider the strategy not relevant. The decreasing curricula CL-DTF-x that start the training with a high degree of TF and then incrementally reduce it to pure FR training partly perform better than the CL-CTF-P strategy and for a few datasets even substantially better than the baseline. We also proposed and studied increasing curricula CL-ITF-x that start the training with no or a low degree of TF, which is then incrementally increased over the course of the training. We observe that these strategies consistently outperform not only the baseline strategies, but all other tested curriculum learning strategies as well. However, it was not foreseeable when this would be the case, making their application not suitable for new datasets without a lot of experimentation and tuning.

**Training length (RQ3).** Choosing an improper teaching strategy can result in an early convergence on a high level of generalization error, e.g., TF strategy for Mackey-Glass, Thomas, and Rössler. Models that yield better performance typically train for more iterations (cp. Tab. III and IV). However, a longer training may not necessarily yield a better performance, e.g., FR vs. TF for Lorenz. When considering the best-performing CL-ITF-x strategies compared to the best-performing baseline strategy, we observe moderately increased training iterations for some datasets, i.e., Thomas, Rössler, Hyper Rössler, Lorenz, but also decreased training iterations for other datasets, i.e., Mackey Glass and Lorenz'96. To better understand whether the longer training is the true reason for the better-performing CL-ITF-x models, we compared their performance when only trained for as many iterations as the baseline model and still observe superior performance over it. In conclusion, we observe that the CL-ITF-x strategies facilitate a robust training, reaching a better generalizing model in a comparable training time on all six tested datasets.

**Prediction stability (RQ4).** We evaluated the generalization as NRMSE for all models trained with the different training strategies per dataset, while forecasting a dataset-specific horizon of $1\,$LT. However, this metric reflects only an average. When we strive for higher model performance on a multi-step prediction task, we often aim for a longer prediction horizon at an acceptable error. To compare prediction stability, we report the NRMSE metric separately, solely computed on the last 10% of the 1 LT horizon. Additionally, we computed how many LTs per datasets can be predicted before falling below an $R^2$ of 0.9. We found that the CL-ITF-x strategies yielded the lowest NRMSE of the last 10% predicted values across all datasets and even more promising that these strategies facilitated the longest prediction horizon without falling below $R^2 = 0.9$. We conclude that the CL-ITF-x strategies may help models find more stable minima in training, substantially improving their long-term forecasting performance.

**Curriculum parametrization (RQ5).** Initially, we evaluated curriculum learning strategies with a variety of different transition functions and individual start and end TF rate (cp. *exploratory* experiments). In these experiments, we observed high prediction performance of the CL-ITF-x strategies, but with a diverse dataset-specific best-performing curriculum, meaning that the application of these strategies for new datasets would have necessitated an extensive hyper-parameter search. In a second set of *essential* experiments, we therefore explored whether we could identify curricula with less parametrization and a similar performance. We found those by using a linear transition that is solely configured by a single parameter Ł that determines the pace with which the TF increases or decreases over the course of the training. We found that these curricula were not only comparable to the previous transition functions and their parametrization, but performed better for all four datasets that we evaluated in both experimental sets and yielded the best-performing model across all six datasets in the second experiment.

**Iteration scale curriculum (RQ6).** Having the CL-ITF-x strategies outperforming the CL-DTF-x strategies leads to rethinking the hitherto common intuition of supporting the early phases of training by TF and moving towards FR in the later stages of training. Rather, we hypothesize that this lures the model into regions of only seemingly valid and stable minima in which they stay even during the more FR-heavy epochs, resulting in a premature termination of the training. This hypothesis is supported by the resulting loss curves for the respective experiments. For example, for the increasing curriculum case, the training duration and the course of the $\epsilon$ (c.p., Fig. 8) show that a training that starts off with FR can enable the model to train much longer in the pure TF zone later. This is compared to the case where solely TF is used. The difference between the two CL-ITF-x strategies is how the prescribed amount of TF is distributed across the prediction steps of one training iteration (aka epoch). While the CL-ITF-D strategy distributes them as one cohesive sequence, the CL-ITF-P strategy distributes them randomly across the training sequence. We found that in the *essential* experiments with the linear transition, the CL-ITF-P strategy performed overall best for four

of the six datasets and would have also been a good choice with a substantial gain over the best-performing baseline training strategy for the other two datasets. In conclusion, we observe that the CL-ITF-P strategy trains models that yield 16–81% higher performance than a conventional training with FR or TF. Apart from that, the *essential* results do not lead to a clear conclusion whether to use CL-ITF-P or CL-ITF-D in a given case. The above-mentioned most obvious difference in the distribution of TF steps firstly may lead to a more coherent backpropagation in the deterministic variant, but it also results in a different behavior regarding maximum number of consecutive FR steps (TF-gap) for a given $\epsilon$. Having the same curriculum function applied for CL-ITF-P and CL-ITF-D makes the TF-gap decrease much faster in the early training stage for the probabilistic variant. Further, it changes the TF-gap in a logarithmic, rather than a linear fashion as for CL-ITF-D. This difference cannot be compensated by parametrizing the curriculum length, demonstrating the need for both strategies. Plus, this only affects the mean TF-gap produced by CL-ITF-P, which has a variance of $\frac{1-\epsilon}{\epsilon^2}$ due to its geometric distribution. Therefore, the TF-gap also varies a lot in the early training stage. In the context of the result Tables VI and VII, we see the probabilistic and the deterministic type of CL-ITF-x can exhibit disfunctions in some specific cases. More thorough investigation of such cases is required to further improve CL strategies, making them more generally applicable and fully identifying their limitations.

**Limitations of this work.** Our observations allow us to draw conclusions regarding appropriate curricula for the training of sequence-to-sequence RNN on continuous time series data. More precisely, the data in our study originates from dynamical systems that predominantly impose chaotic behavior. Consequently, all observations made here are currently limited to these kind of models and data. We acknowledge that more research is necessary to clarify several uncertain points. First, regarding the question why an increasing curriculum tends to find more stable minima throughout all studied datasets. Leading to the question, what determines a proper curriculum and parametrization. A first step could be to shift the focus on the baseline strategies again to extract primary conditions under which trainings get unstable. Therefore, a closer look at the weights and behavior of the model gradients during training, the statistics of the gradient of the processed time series and the used sampling rate is required at least. Finally, a more thorough investigation on empirical real-world data improving on the early and inconclusive results on the Santa Fe dataset (cp. Tab. VI) is needed to reveal the capabilities and limitations of different CL strategies in this regard.

**Application in other domains.** In this paper we focus exclusively on the field of time series forecasting. However, the proposed strategies could easily be applied to ML models exhibiting autoregressive predictions trained for other domains. For example, sequence-to-sequence RNNs and transformers for Automatic Machine Translation (AMT). As mentioned before, scheduled sampling, that is included in the CL-DTF-x strategies, was originally proposed, and successfully used to improve the training of RNNs for NLP, but, on the other hand, performed rather poorly for time series forecasting in our experiments. Thus, we cannot draw any conclusions about the usefulness of CL-ITF-x strategies in other domains based on our results either. Therefore, evaluating the effectiveness of increasing curricula in other domains remains a task for future work and, at this point, would go beyond the focus of our study.

## 7 Conclusions

While training encoder-decoder RNNs to forecast time series data, strategies like TF produce a discrepancy between training and inference mode, i.e., the exposure bias. Curriculum learning strategies like scheduled sampling or training directly in FR mode are used to reduce this effect. We run an extensive series of experiments using six chaotic dynamical systems as benchmark and observed that not for all of them the exposure bias yields a crucial problem. We see that even an FR training that is per definition free of exposure bias may not be optimal. In fact, we observed that for half of our datasets, TF performs better, while for the other half, FR yields better results. With the focus on curriculum learning strategies like scheduled sampling, we proposed two novel strategies and found that those yield models that consistently outperform the best-performing baseline model by a wide margin of 15–80% NRMSE in a multi-step prediction scenario over one Lyapunov time. We found that these models are robust in their prediction, allowing to forecast longer horizons with a higher performance. We found it sufficient to parametrize the strategy with a single additional parameter adopting the pace of the curriculum.

**Acknowledgments**

This research was funded by the project P2018-02-001 "DeepTurb – Deep Learning in and of Turbulence" of the Carl Zeiss Foundation; the German Federal Ministry for the Environment, Nature Conservation, Nuclear Safety and Consumer Protection (BMUV) grant: 67KI2086A; and the German Ministry of Education and Research (BMBF) grant: 01IS20062B.

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

## A    Training and Validation Loss Curves

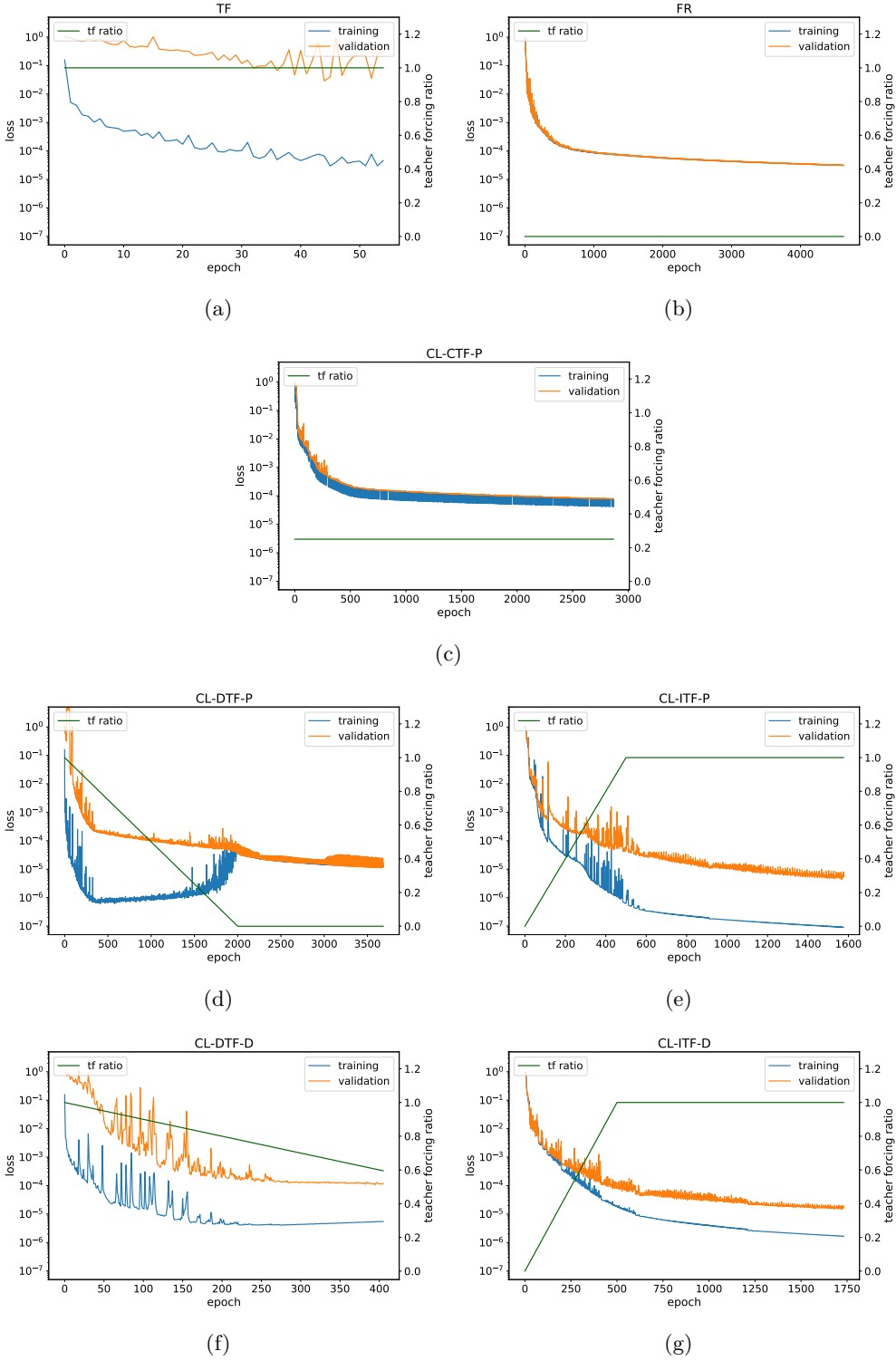

Figure 13: Training and validation loss for Mackey-Glass

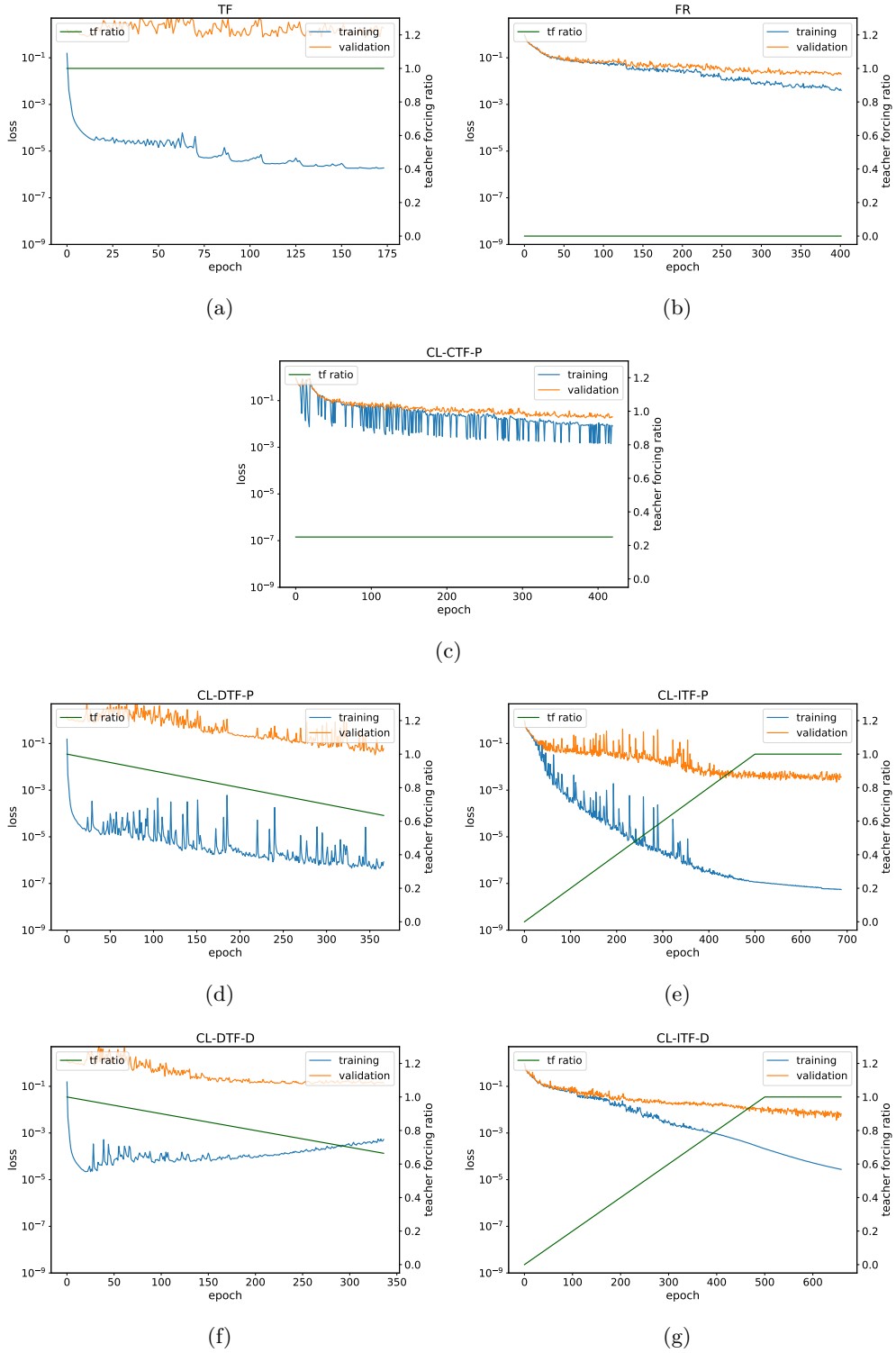

Figure 14: Training and validation loss for Thomas

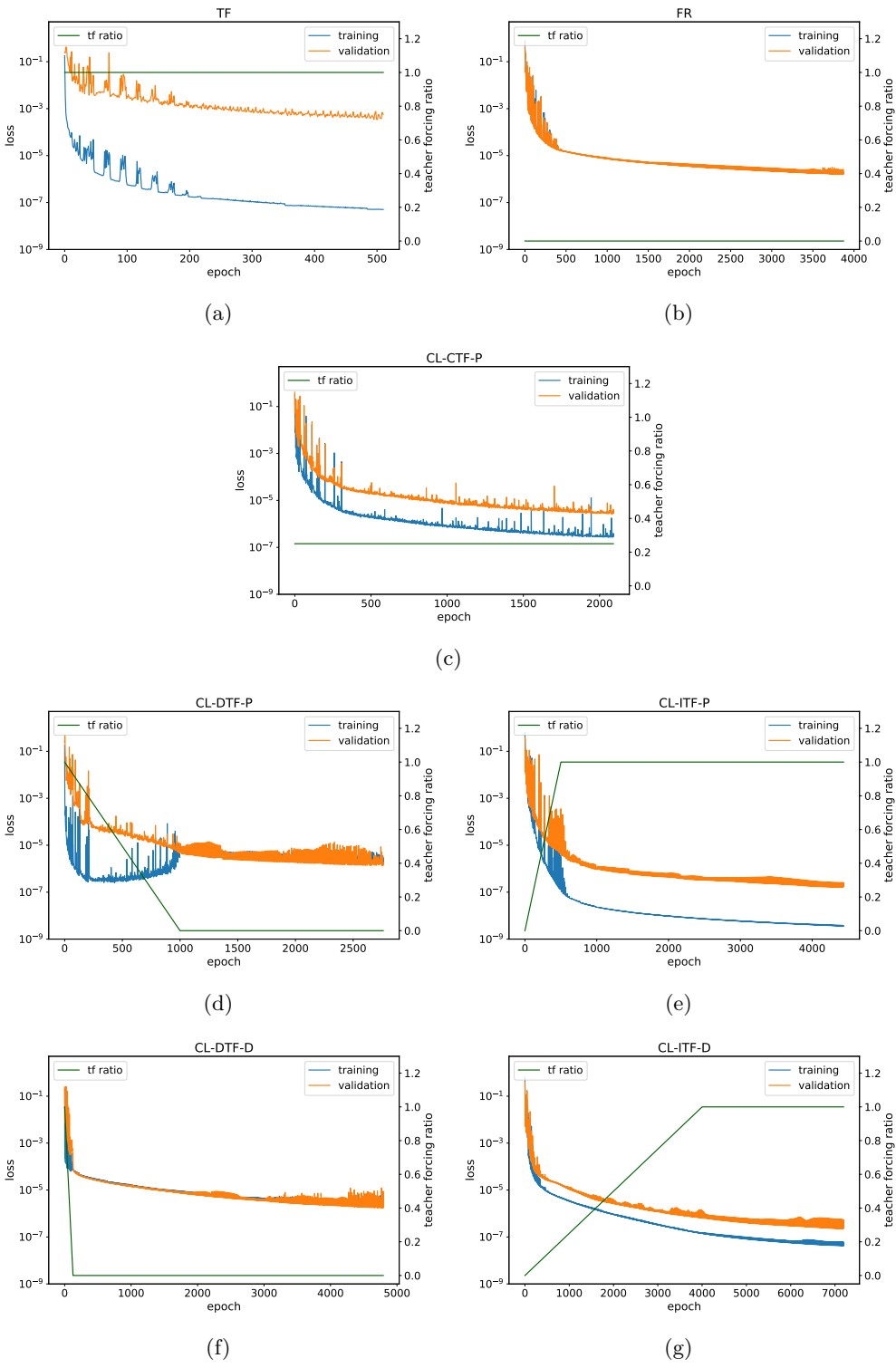

Figure 15: Training and validation loss for Rössler

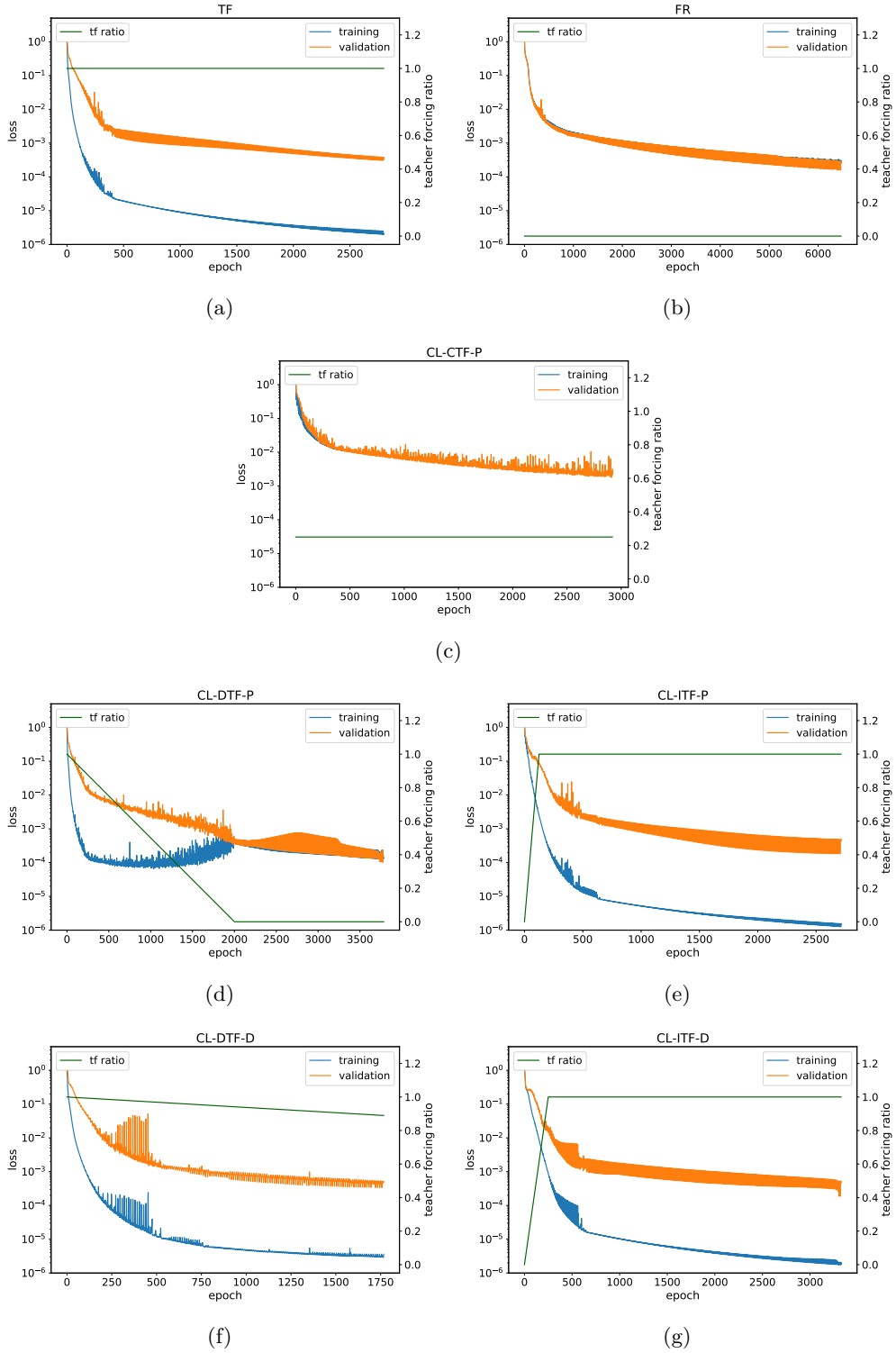

Figure 16: Training and validation loss for Hyper-Rössler

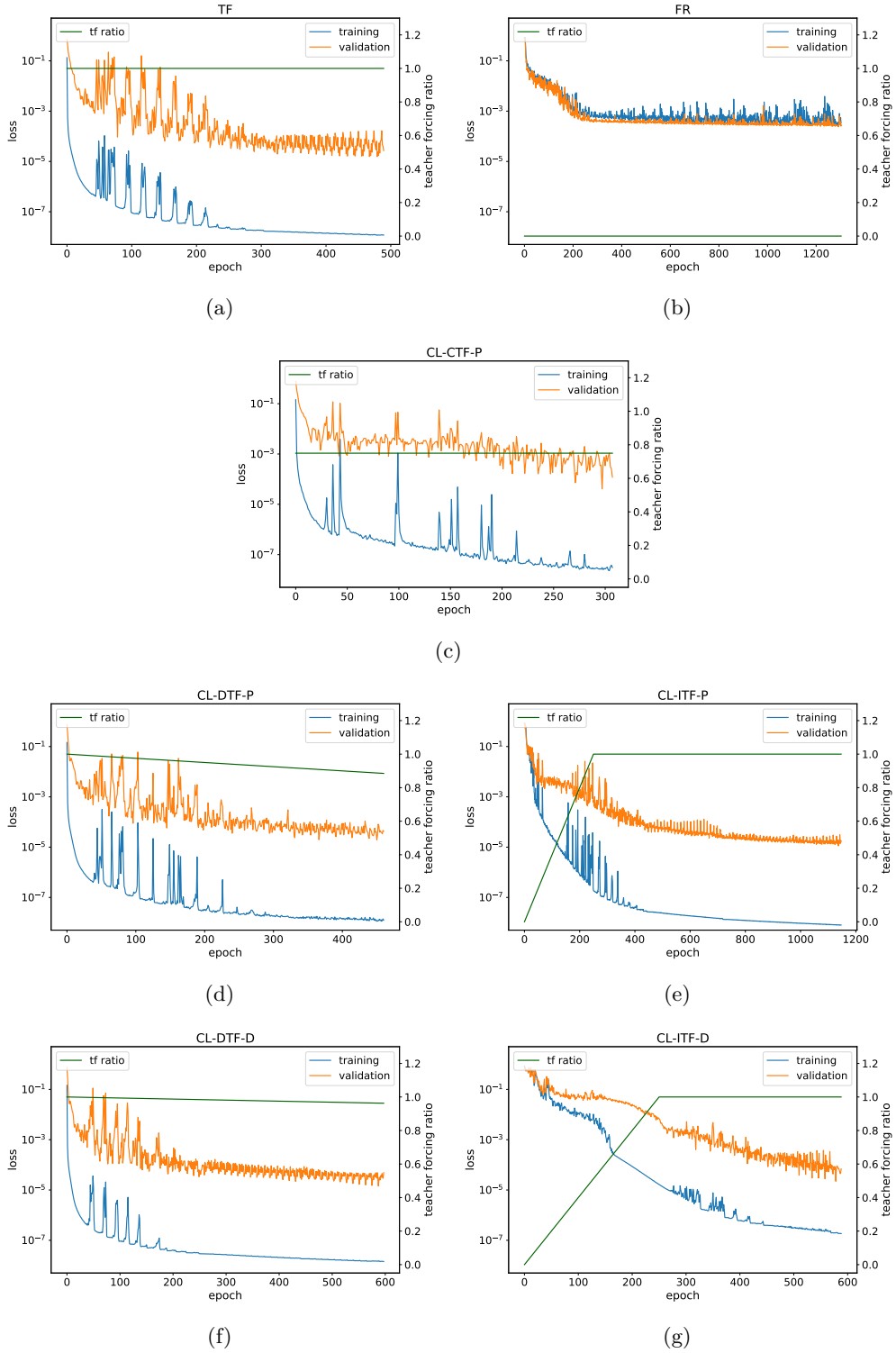

Figure 17: Training and validation loss for Lorenz

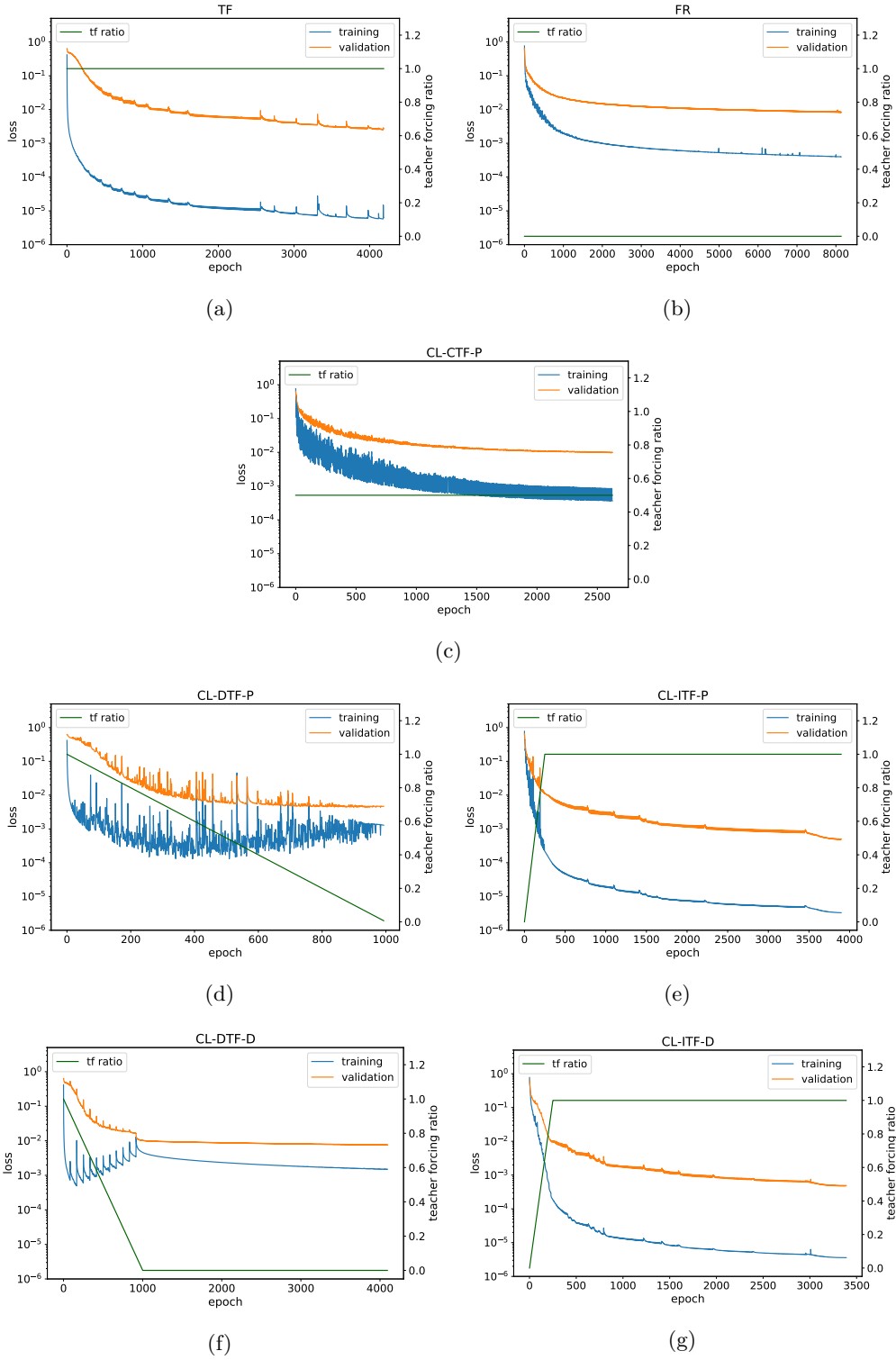

Figure 18: Training and validation loss for Lorenz'96

## B    CL-ITF-P With and Without LR Scheduler

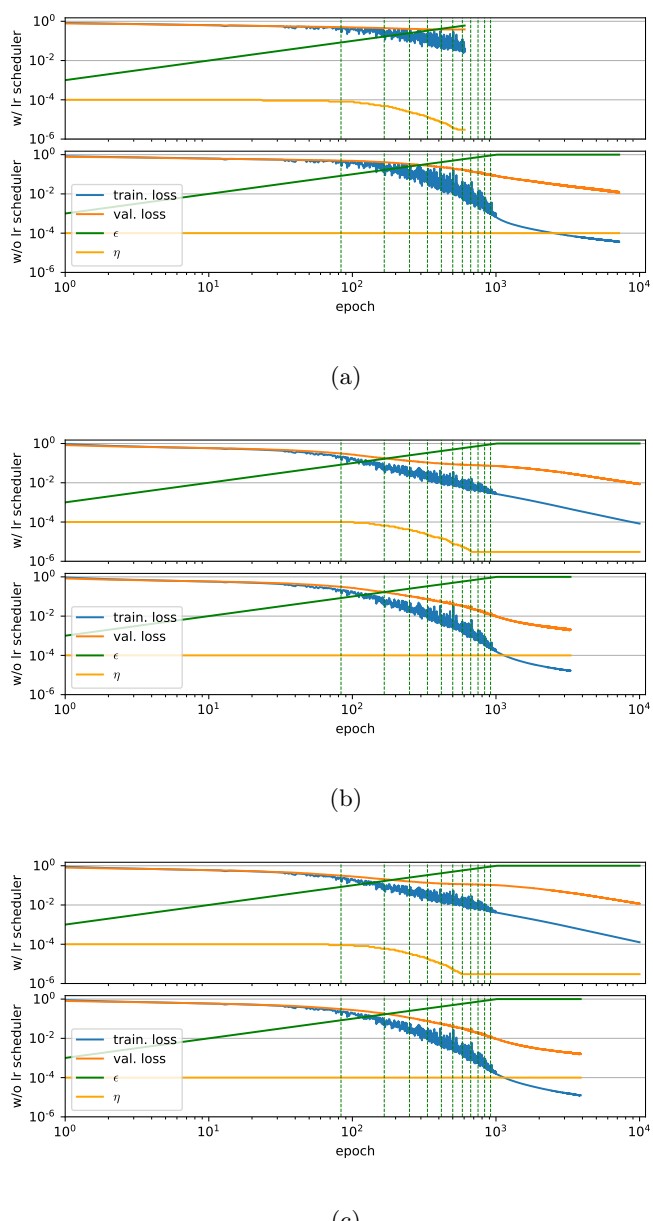

(a)

(b)

(c)

Figure 19: Course of the training and validation loss for the basic RNN (a), LSTM (b) and GRU (c) on Lorenz'96 data with and without using a learning rate scheduler.

## C  Different RNN Models

Table VIII: Forecasting performance of the vanilla RNN on four different chaotic datasets

| System | Strategy | Ł | Epochs | NRMSE ↓ | Rel. impr. ↑ |
|---|---|---|---|---|---|
| Thomas | FR | – | 51 | 0.48117 | – |
| | TF | – | 249 | 0.41274 | – |
| | CL-ITF-P | 1 000 | 266 | **0.17955** | 56.50% |
| | CL-ITF-D | 250 | 183 | 0.25659 | 37.83% |
| Rössler | FR | – | 2 292 | 0.00747 | – |
| | TF | – | 1 | 0.50174 | – |
| | CL-ITF-P | 500 | 2 063 | **0.00283** | 62.12% |
| | CL-ITF-D | 1 000 | 3 075 | 0.00319 | 57.30% |
| Lorenz | FR | – | 506 | 0.11389 | – |
| | TF | – | 572 | 0.00913 | – |
| | CL-ITF-P | 1 000 | 746 | 0.00603 | 33.95% |
| | CL-ITF-D | 125 | 782 | **0.00378** | 58.60% |
| Lorenz'96 | FR | – | 1 710 | 0.31002 | – |
| | TF | – | 637 | 0.57870 | – |
| | CL-ITF-P | 1 000 | 505 | 0.56872 | −83.45% |
| | CL-ITF-D | 1 000 | 6 573 | **0.07663** | 75.28% |

Table IX: Forecasting performance of the LSTM on four different chaotic datasets.

| System | Strategy | Ł | Epochs | NRMSE ↓ | Rel. impr. ↑ |
|---|---|---|---|---|---|
| Thomas | FR | – | 45 | 0.43698 | – |
| | TF | – | 818 | 0.05265 | – |
| | CL-ITF-P | 250 | 758 | 0.01892 | 64.06% |
| | CL-ITF-D | 125 | 859 | **0.01181** | 77.57% |
| Rössler | FR | – | 2 417 | 0.00210 | – |
| | TF | – | 1 650 | 0.00139 | – |
| | CL-ITF-P | 1 000 | 3 426 | **0.00063** | 54.68% |
| | CL-ITF-D | 250 | 2 367 | 0.00085 | 38.85% |
| Lorenz | FR | – | 1 154 | 0.06526 | – |
| | TF | – | 398 | 0.00169 | – |
| | CL-ITF-P | 250 | 806 | **0.00075** | 55.62% |
| | CL-ITF-D | 31 | 419 | 0.00125 | 26.04% |
| Lorenz'96 | FR | – | 4 019 | 0.13010 | – |
| | TF | – | 3 721 | 0.22757 | – |
| | CL-ITF-P | 500 | 9 164 | **0.03935** | 69.75% |
| | CL-ITF-D | 62 | 4 509 | 0.06855 | 47.31% |

Table X: Forecasting performance of the unitary evolution RNN on four different chaotic datasets.

| System | Strategy | Ł | Epochs | NRMSE ↓ | Rel. impr. ↑ |
|---|---|---|---|---|---|
| Thomas | FR | – | 354 | 0.74991 | – |
| | TF | – | 53 | 0.85679 | – |
| | CL-ITF-P | 64 000 | 359 | **0.61889** | 17.47% |
| | CL-ITF-D | 64 000 | 340 | 0.73331 | 2.21% |
| Rössler | FR | – | 911 | 0.03726 | – |
| | TF | – | 1 287 | 0.03317 | – |
| | CL-ITF-P | 1 000 | 1 126 | 0.02473 | 25.44% |
| | CL-ITF-D | 500 | 1 402 | **0.02420** | 27.04% |
| Lorenz | FR | – | 758 | 0.13021 | – |
| | TF | – | 4 | 0.96357 | – |
| | CL-ITF-P | 1 000 | 758 | **0.03098** | 76.21% |
| | CL-ITF-D | 500 | 1 096 | 0.02897 | 77.75% |
| Lorenz'96 | FR | – | 1 043 | 0.26359 | – |
| | TF | – | 398 | 0.61159 | – |
| | CL-ITF-P | 2 000 | 697 | 0.26253 | 0.4% |
| | CL-ITF-D | 2 000 | 1 335 | **0.18483** | 29.88% |

Table XI: Forecasting performance of the Lipschitz RNN on four different chaotic datasets.

| System | Strategy | Ł | Epochs | NRMSE ↓ | Rel. impr. ↑ |
|---|---|---|---|---|---|
| Thomas | FR | – | 879 | 0.11697 | – |
| | TF | – | 928 | 0.05714 | – |
| | CL-ITF-P | 250 | 879 | 0.05237 | 8.35% |
| | CL-ITF-D | 500 | 1 191 | **0.03926** | 31.29% |
| Rössler | FR | – | 3 704 | 0.27122 | – |
| | TF | – | 79 | 0.90844 | – |
| | CL-ITF-P | 250 | 2 304 | 0.05185 | 80.88% |
| | CL-ITF-D | 500 | 3 149 | **0.04409** | 83.74% |
| Lorenz | FR | – | 1 298 | 0.06675 | – |
| | TF | – | 625 | 0.02830 | – |
| | CL-ITF-P | 62 | 914 | **0.01732** | 38.80% |
| | CL-ITF-D | 250 | 686 | 0.02417 | 14.59% |
| Lorenz'96 | FR | – | 489 | 0.62710 | – |
| | TF | – | 469 | 0.65122 | – |
| | CL-ITF-P | 1 000 | 477 | **0.62170** | 0.86% |
| | CL-ITF-D | 4 000 | 489 | 0.62581 | 0.21% |

