# OpenReview forum: "Flipped Classroom: Effective Teaching for Time Series Forecasting"
_TMLR — Accepted by TMLR_

### Review · Reviewer_EFQM · 2022-07-19

**Summary Of Contributions:**

This paper aims to study methods that can alleviate exposure bias while training sequence-to-sequence models using GRU networks, specifically for the task of time series forecasting. The authors focus on combining the two most commonly used training strategies - Teacher Forcing(TF) & Free Running(FR) using various schemes. TF always uses actual input during the training phase, running a higher risk of exposure bias during the inference stage. FR on the other hand always uses the output from the model while training, which usually results in longer training periods. The authors propose a number of ways to combine TF & FR (called curriculums) to take advantage of both the methods while tackling exposure bias. The curriculums are framed by using both deterministic & probabilistic strategies that switch between TF & FR. The switches can happen after a certain iteration of the training stage (training scale) or after a certain timestamp of a given iteration (iteration scale).

Evaluation (using NRMSE & R2) at various parts of the prediction horizon, on dynamical systems with varying degrees of chaos suggests mixed results
But the suggested curriculums outperform the pure TF or FR settings. Also curriculums starting out with FR and increasing TF are observed to be dominant & most stable.


**Requested Changes:**

There has been no or only minimal attempts to explain some of the counterintuitive results observed:
1. Superior performance of the CL-ITF-X strategies against the common intuition of strategies with TF heavy early epochs. There is mention of a hypothesis but can this be expanded further?
2. Non-trivial performance decrease of the CL-ITF-P strategy using a vanilla RNN on Lorenz’96.
3. Underwhelming performance of the CL-ITF-X strategies on the Santa Fe data set.


**Strengths And Weaknesses:**

Strengths:
1. Experiments that clearly show exposure bias exists in chaotic time series forecasting, especially when pure TF strategy is used
2. Systematic evaluation setup, with metrics to measure deterioration of forecast quality over prediction horizon

Weakness:
1. Directionally the CL-ITF-X strategies seem to outperform the baseline & other curriculums but there is no analysis on why this is the case. (Some hypotheses & future directions are mentioned in the limitations section.)
2. All results are empirical, limiting generalizability.

---

> ### Author Response · Authors · 2022-08-22
> **Response to Request 1.1**
>
> R1.1: Superior performance of the CL-ITF-X strategies against the common intuition of strategies with TF heavy early epochs. There is mention of a hypothesis but can this be expanded further?
>
> A1.1: Thank you for your comment! We aim to support our hypothesis further by pointing at the newly added loss curves. We also added additional discussion to highlight this connection in the RQ6 part of the Discussion section:
> “[…], we hypothesize that this lures the model into regions of only seemingly valid and stable minima in which they stay even during the more FR heavy epochs resulting in a premature termination of the training. This hypothesis is supported by the resulting loss curves for the respective experiments. For example, for the increasing curriculum case, a look at the loss curve and the respective course of \epsilon (cp. Fig. 8) shows that a training that starts off with FR can enable the model to train much longer in the pure TF zone later. This is compared to the case where solely TF is used.”

---

> ### Author Response · Authors · 2022-08-22
> **Response to Request 1.2**
>
> R1.2: Non-trivial performance decrease of the CL-ITF-P strategy using a vanilla RNN on Lorenz’96.
>
> A1.2: We shed more light onto this case now. We found that the interaction of the RNN, Lorenz96 and the probabilistic nature of the CL strategy led to a bad performance due to the used learning rate scheduler. We discuss the matter in more detail in the updated Section 5.7:
> “In our experiments on the Lorenz’96 data, the performance of the encoder-decoder RNN remains an outlier. Especially, when we consider that for the deterministic variant CL-ITF-D there is a clear NRMSE improvement by 75.28%. We use this setup to take a closer look on the training gradients and process. In Figure 11a, we plot the mean of the decoder RNN gradient norm over the training epochs for the CL-ITF-P and the CL-ITF-D case. We also add the teacher forcing probability \epsilon and the learning rate \eta to the plot. For comparison with the two successful setups, we also give an overview of the gradient norm of the GRU and the LSTM based model under the same conditions (cp. Figs 11c and 11b).
>
> For all models, we observe an oscillation of the gradient norm during CL-ITF-P training. CL-ITF-D avoids this behavior with a constant iteration scale curriculum that keeps the TF-FR-pattern fix for several epochs between the changes. The crucial epochs are indicated by green dashed vertical lines. In contrast, CL-ITF-P provides a probabilistic iteration scale curriculum that leads to intra epoch changes. The resulting variation in the gradient norm only stops when the training scale curriculum hits the final \epsilon = 1. Preceding that, the oscillating gradient norm originates from an unsteady training loss. Most of the time the unsteady loss can be compensated due to the increasing amount of TF over the epochs. For the Lorenz’96 data, however, it seems the unsteady loss together with the used scheduler leads to a rapid decrease of the learning rate. This effect is observable during the curriculum training of the basic RNN, the LSTM and the GRU. The latter two models seem to handle the resulting minimal learning rate of 3e−6 much better and manage to continue their training successfully.
>
> To get an impression of the RNN’s behavior without the unwanted learning rate drop we conducted the same CL-ITF-P experiments again without applying any learning rate scheduler. This time we plot the training loss, that exhibits the same oscillating behavior as the gradient norm, together with the validation loss, to see if the change leads to improved loss values. The plot in Figure 12 shows that without the unwanted learning rate drop the basic RNN remains stable the transient phase towards \epsilon = 1 and ends up with a validation loss of 0.012 instead of 0.373. For the more sophisticated RNNs, we see clearly improved validation losses after much less epochs as well. The corresponding plots are shown in appendix A.2.”

---

> ### Author Response · Authors · 2022-08-22
> **Response to Request 1.3**
>
> R1.3: Underwhelming performance of the CL-ITF-X strategies on the Santa Fe data set.
>
> A1.3: To get a better intuition about the Santa Fe performance, we added another set of experiments where we forecast 200 steps. It shows that the increase of the tasks difficulty in this case also leads to a clearer separation of the different strategies’ performances. We added a more exhaustive analysis of the CL-ITF-D training behavior as well:
> “When increasing the prediction horizon from 20 to 200 steps, we observe an increased relative improvement of 26.27% for CL-ITF-P. At the same time, we observe a relative decrease of −5.89% for CL-ITF-D. To investigate the radical difference between these results, we study \epsilon in relation to the appearance of TF steps in the respective training curriculum. Figure 10 shows two different probabilities along the training epochs. There is the TF probability \epsilon that is directly determined by the curriculum function and p_{\leq200} which denotes the probability of a training sample to contain at least one TF step. It is computed as p = $formula (see document) for the probabilistic strategy. For CL-ITF-D this probability is either 0 or 1 meaning that it only has an effect on training epochs >= 160. If we would use the same Ł as for CL-ITF-P here, the training would converge after 208 epochs just as in the plain FR case. CL-ITF-P though, due to its probabilistic nature, can sporadically induce TF steps for some of the batches in the earlier phase of training and thus provides a much smoother transition towards TF phase. We argue that this smoothness is crucial for the given task, which appears very sensitive to TF training steps.”

---

### Review · Reviewer_9Rk6 · 2022-07-21

**Summary Of Contributions:**

This article studies the exposure bias in chaotic time series forecasting. Exposure bias, which is commonly available in sequence prediction for many NLP tasks, has been widely explored in literature. Scheduled sampling (Bengio et al., 2015), which is one of most effective solutions to migrate this issue, has been adopted by authors. Their main contribution is two folds:

demonstrate that exposure bias is also a big issue in chaotic time series.
provide extensive empirical experiments of these training strategies on chaotic time series.



**Requested Changes:**

The revised version has addressed most of my concerns.
I'd like to give an acceptance.

**Strengths And Weaknesses:**

Strengths:

demonstrate exposure bias is the issue in chaotic time series.

provide extensive experiments to study different training strategies showing that the probabilistic iteration scale curriculum works better.

Weaknesses:

Limited novelty. These training strategies are highly similar to ones from scheduled sampling (Bengio et al., 2015).

Introduced more free hyper-parameter needed to tune.

---

> ### Author Response · Authors · 2022-08-22
> **Thank you**
>
> Thank you for your positive review.

---

### Review · Reviewer_EVtJ · 2022-08-07

**Summary Of Contributions:**

This paper studies the risk of exposure bias in training sequential models with teacher-forcing vs free-running, i.e.,

(a) Teacher-forcing (TF): Each timestep uses the correct input from the teacher
(b) Free-running (FR): Each timesteps uses the model prediction from the previous step as the input.

Teacher-forcing leads to faster training convergence but risks exposure bias since the teacher always provides the correct inputs during the training stage, the model is not exposed to its predictions as input from the previous stages. Since there is no teacher availability during the inference stage, the model risks exposure bias.

Free-running removes the exposure bias during the inference stage as the model has been trained with its predictions from the past (and this is inline with the test time setup as the teacher is not available during the inference stage)


This paper's proposal is to provide a set of curriculum that decides what fraction of the input timesteps uses the teacher inputs and what fraction uses the model predictions from previous stages. In contrast to previous works, this paper proposes curriculum that switches between TF and FR on two different scales:
(i) depending on training iteration ( ex. start with TF in the initial iterations and then decay  to use  FR )
(ii) within a training iteration ( ex. initial timesteps use TF while later timesteps use FR)

Paper evaluates the proposed curriculum learning strategies on various dynamical systems including chaotic as well as periodic behavior.


**Requested Changes:**

Given the above listed weaknesses, I do think the paper merits acceptance mainly due to the fact that the work aggregates various CL strategies and compares (with some reservations on various architectures) on a series of meaningful dynamical systems that simulate time-series forecasting. Even though the models are restricted to GRUs mainly, it still shows a simple scheme can be meaningfully adopted to yield non-trivial results.

To strengthen their work,  I would recommend the authors to keep the same model structure but use couple of RNNs ( say LSTMs, ODE-RNNs, Unitary RNNs, etc. ) and show that their proposed scheme works across architectures.

Further more, I had commented in previous submission w.r.t. general applicability of the scheme extending beyond chaotic systems. I acknowledge that the authors included the comparison on the periodic dynamical systems. It would be interesting to see if the similar curriculum extends to other domains such as seq-to-seq text generation or audio generation. Or the authors to comment as to what might be the limitations of extending the proposed scheme to other domains.

**Strengths And Weaknesses:**


Strengths:
-----------

Simplicity of the proposed curriculum specially the probabilistic strategies that allow switching between teacher forcing or free running.

Extensive empirical evaluation on different chaotic dynamical systems.


Weaknesses (Earlier version):
-----------

Although the proposed method is simple, and it sheds some light on the different curriculum learning strategies, the fact remains that its an incremental update on the proposed methods in the literature.

The paper mainly focuses on GRU based auto-encoders and uses some ablations to study the impact of the proposed scheme on LSTMs.

-----

Given the above listed weaknesses, I do think the paper merits acceptance mainly due to the fact that the work aggregates various CL strategies and compares (with some reservations) on a series of meaningful dynamical systems that simulate time-series forecasting. Even though the models are restricted to GRUs mainly, it still shows a simple scheme can be meaningfully adopted to yield non-trivial results.

---

> ### Author Response · Authors · 2022-08-22
> **Response to Request 3.1**
>
> R3.1: To strengthen their work, I would recommend the authors to keep the same model structure but use couple of RNNs (say LSTMs, ODE-RNNs, Unitary RNNs, etc. ) and show that their proposed scheme works across architectures.
>
> A3.1: Thank you for your positive feedback and constructive impulses. We added models based on Unitary RNNs (uRNNs) and Lipschitz RNNs (LRNNs) to the list and present their results together with the vanilla RNN and LSTM in a joint table in the Results section. We also updated the discussion of results accordingly:
> “The uRNN and the LRNN do not adapt their weight matrices directly while training but rather update several structured building block matrices to construct the actual weight matrices. We observe improvements from 0.4 to 77.75% and from 0.21 to 83.74% for the two architectures respectively. We provide more detailed information about these experiments in appendix A.2.”

---

> ### Author Response · Authors · 2022-08-22
> **Response to Request 3.2**
>
> R3.2: Furthermore, I had commented in previous submission w.r.t. general applicability of the scheme extending beyond chaotic systems. I acknowledge that the authors included the comparison on the periodic dynamical systems. It would be interesting to see if the similar curriculum extends to other domains such as seq-to-seq text generation or audio generation. Or the authors to comment as to what might be the limitations of extending the proposed scheme to other domains.
>
> A3.2: We do not think that our results allow us to expect the effectiveness of the tested strategies in other fields in general. We added another paragraph to the Discussion section to deal with this uncertainty:
> “In this paper, we focus exclusively on the field of time series forecasting. In principle the proposed strategies could easily be applied to ML models exhibiting autoregressive predictions trained for other domains. For example, sequence-to-sequence RNNs and Transformers for Automatic Machine Translation (AMT). As mentioned before, scheduled sampling, that is included in the CL-DTF-x strategies, was originally proposed, and successfully used to improve the training of RNNs for NLP but, on the other hand, performed rather poorly for time series forecasting in our experiments. Thus, we cannot draw any conclusions about the usefulness of CL-ITF-x strategies in other domains based on our results either. Therefore, evaluating the effectiveness of increasing curricula in other domains remains a task for future work at this point and would go beyond the focus of our study.”

---

### Author Response · Authors · 2022-08-22
**Dear Reviewers**

thank you again, for your precise feedback for our paper. We did our best to address all requests accordingly and uploaded the updated version of the PDF. You will find the new or changed text passages highlighted in color.

Best Regards,

The Authors

---

### Decision · Action_Editors · 2022-09-12

**Recommendation:** Accept as is

**Comment:**

This paper examines teaching strategies for sequential data, highlighting challenges with teacher forcing and with free running, and introduces a curriculum learning approach that is the subject of a thorough empirical investigation. Relative to the last submission, this version has significantly improved the narrative and clarity of argument, and the overall message is now clear and well supported by evidence. There remains some hesitation about the novelty and potential impact of the results, as the curriculum learning schedule is similar to prior work. However, the relatively small modifications suggested here do seem to have significant impact, and the empirical analysis is thorough. Some reviewers do suggest a broader set of experiments, with additional architectures etc., but given the breadth of the analysis already undertaken, all reviewers and I agree that the current version merits acceptance.

---

> ### Author Response · Authors · 2022-10-04
> **Camera-Ready Version**
>
> Dear reviewers and AE
>
> Thank you for the time and effort invested in the review process. It helped us to improved the quality of the paper. We just uploaded the de-anonymized camera-ready version.
>
> Best Regards,
>
> The Authors